# Magneto-responsive chain-like arrangements of size-tuned and cobalt-doped ferrites derived from silica-encapsulated precursors
Maria Weißpflog [ORCID] ✉, Julia Kabelitz [ORCID] & Birgit Hankiewicz [ORCID] ✉

Anisotropic materials, such as intracellular nanochains in magnetotactic bacteria, exhibit significant potential in biomedicine and technology due to their magnetic, direction-dependent properties. However, their synthesis is limited due to scalability and purification issues. Here, we present an alternative route to bioinspired magneto-responsive nanostructures, specifically chain-like arrangements composed of truncated cubic cobalt-doped ferrite particles. The magnetic nanoparticles are synthesized in an eco-friendly manner *via* coprecipitation and hydrothermal conversion of silica-coated nanorods with varying shell thicknesses. A distinct relationship emerged between the cobalt-to-iron ratio, nanoparticle dimensions, and mass magnetization, revealing that these parameters increase with silica shell thickness and reaction temperature. When an external magnetic field is applied, the randomly distributed particles align themselves into nanochains, facilitating the determination of the number of particles in both parallel and perpendicular orientations, as calculated from Small-Angle X-ray Scattering analysis. It is observed that the cluster numbers vary in comparison to the dipole-dipole interaction energy and particle size due to the formation of these chain-like structures and bundles. A critical evaluation of the characteristics of both individual particles and chains summarizes their suitability for biomedical applications.

It has already been demonstrated in nature that anisotropic materials can serve as an extraordinary tool due to their direction-dependent properties when examining intracellular nanochains found in magnetotactic bacteria[1–3]. These structures consist of isotropic building units formed from ferrimagnetic nanoparticles, which enable navigation in a minimal magnetic field $H < 0.067$ mT (*e.g.*, the Earth's magnetic field)[4–6] and open discussions on their potential for medical use[2,7,8]. Doping with cobalt ions has also been investigated to study the effects of increased magnetic anisotropy on the magnetic behavior of the magnetosomes[9]. However, it should be noted that their application remains controversial, primarily because of the limited scalability of cultivation and purification[10]. Nonetheless, this research area is highly prominent, and the imitation of these nanostructured systems has aroused considerable interest due to their diverse applications in biomedicine (medical therapies[11,12], directed cell growth[13], cell separation[14], hyperthermia treatment[15], etc.) and technology (sensors[16], nanorobotics[17], telecommunication[18], catalysis[19], microfluidics[20], etc.).

Nanochain assemblies can be achieved by various methods, such as spontaneous self-assembly[21–23], chemical-directed bonding[24,25], forth-directed alignment[26–29], and others[30,31]. For example, Nandakumaran et al. demonstrated a reversible process by adjusting the length of self-assembled nanostructures through changes in the magnetic field strength[22]. However, these structures exhibit less control during synthesis and less stability without a magnetic field, as the primary force originates from the dipole-dipole interaction of the magnetic particles[32]. This depends on factors such as volume, concentration, magnetization, viscosity, and fluidic stability[32–35]. By incorporating various moieties on the particle's surface, molecular interactions such as electrostatic, hydrogen bonding, DNA-mediated, or hydrophobic confinement can be utilized to arrange magnetic nanostructures in a directed way[31,36]. Nonetheless, these interaction-directed methods are mainly combined with the application of a magnetic field during crosslinking (*e.g.*, magnetic-field-assisted Diels-Alder reaction or copper-catalyzed alkyne-azide cycloaddition click reaction) and necessitate

University of Hamburg, Institute of Physical Chemistry, Grindelallee 117, Hamburg, Germany. ✉e-mail: maria.weisspflog@uni-hamburg.de; birgit.hankiewicz@uni-hamburg.de

a more intricate approach[24,25]. The force-directed alignment predominantly necessitates the application of an external magnetic field in conjunction with the subsequent incorporation of a polymer or silicon layer surrounding these structures[26–29]. This additional layer is essential for stabilizing the magnetic nanoparticles (MNPs) within their configuration, owing to their relatively weaker magnetic properties or when removing the external magnetic field[37]. Several studies demonstrate the successful design of chains of varying lengths made from magnetic materials surrounded by silica or oxide layers of different thicknesses to address the stability problem[15,27,38,39]. Moreover, including shells can enhance such a system's biocompatibility[38]. In particular, we would like to refer to works focused on the construction of silica-coated, at least magnetically-responsive nanochain structures, which demonstrate their applications across the various fields mentioned above. This high variation arises from the enhanced magnetic and shape anisotropy, the reversibility through external fields, the increased (rough) surfaces, the magnetic coupling effects, and other properties of the bioinspired magnetic nanochains[20,27,40–43].

The magnetic components of these systems are primarily based on iron oxides such as magnetite ($Fe_3O_4$) or maghemite ($\gamma$-$Fe_2O_3$) due to their various advantages, including size-dependent magnetic properties[44–46], low synthesis costs[47], and biocompatibility[48]. However, they have a low anisotropy constant, low thermal efficiency, or tend to oxidize[49]. Since the magnetic anisotropy of the particles influences hyperthermia efficiency, expressed by specific absorption rates (SAR), an increase can be achieved through modification, *e.g.*, by altering the particle shape (anisotropy of particle shape) or the crystal structure (anisotropy of crystal structure)[50–52]. Using shape-anisotropic $Fe_3O_4$ nanoparticles, such as cubic, octahedral, or star-like, increases the coercivity, saturation magnetization, and SAR[45,49,51,53,54]. The exchange of Fe(II) ions with Co(II) ions in an inverse spinel structure increases the magneto-crystalline anisotropy and, thus, in the case of cobalt ferrite, results in a more stable material against oxidation and exhibits adequate thermal stability compared to magnetite[45,52,55,56]. As a result of higher magneto-crystalline anisotropy, cobalt ferrite particles should already be dominated by the Brownian relaxation mechanism in the high single-digit nanometer range (around 7 nm), compared to magnetite nanoparticles (around 14 nm). The combination of both anisotropy effects can be realized *via* a partial ion exchange of Fe(II) with Co(II) ions in shape-anisotropic iron oxide particles to obtain $Co_xFe_{3-x}O_4$ (x < 1) particles[57]. For example, Shebha Anandhi et al. achieved an increase in SAR by doping magnetite with cobalt to produce $Co_xFe_{3-x}O_4$ within the concentration range of x ≤ 0.5[55]. This also relates to potential applications in medical cancer treatments, where the hyperthermal effect is arguably the most intriguing aspect[58]. Regarding the toxicity of cobalt-doped particles, it should be noted that these particles exhibit a reduced toxicity within the range of 0.2 < x < 0.6 compared to magnetite and cobalt ferrite[59]. A more detailed analysis of toxicity and biodegradability will be addressed in the final section of this work. In conclusion, considering the different relaxation processes and the hyperthermal effect, this work focuses on the synthesis of cobalt-doped ferrite (CF) particles with a lower cobalt amount (x ≤ 0.5) to increase the magneto-crystalline anisotropy and reduce the toxicity as presented in previous work[57].

Monodisperse cobalt-doped ferrite nanoparticles are mainly synthesized through a thermal decomposition process using toxic metal acetylacetonates or metal pentacarbonyls with lower cobalt content. Despite their exceptional SAR of up to 14,686 W $g^{-1}$ Fe, as exemplified for cubical bipyramids, octahedrons, and hexagons by Singh et al. and Demessie et al., these syntheses require harmful solvents and surface-active surfactants, such as oleylamine[60,61]. Therefore, the development of green or aqueous methods for preparing cobalt-doped ferrite is continuously increasing[62]. Novel environmentally friendly approaches utilize extracts from hibiscus, tea, sesame, honey, cardamom, etc., or are conducted using microorganisms and fungi[63]. However, these methods mostly result in particles with a high size distribution and low yields. Microwave-assisted syntheses are also possible, particularly in combination with combustion or hydrothermal methods, which significantly reduce reaction times and thus energy consumption.

However, microwave hybrid synthesis is often limited to small volumes due to the complicated setup and challenges with slow reaction kinetics[64]. In our group, we have specialized in hydrothermal syntheses, which, although requiring specialized equipment, produce highly crystalline particles with excellent composition characteristics from low-cost precursors[57,64]. Due to the multitude of adjustable parameters, these methods enable a wide variety of particle morphologies, even in aqueous systems without the requirement for toxic solvents, surfactants, or precursors[65,66].

Additionally, the synthetic pathway aims to simultaneously arrange the CF particles in nanochain-like structures by implementing a silica shell around the iron precursors during the synthesis. Therefore, a modification of the well-known precursor-based hydrothermal reaction was carried out. For this purpose, the spindle-like precursor $\beta$-Fe(III)OOH (akaganeite) was first produced from Fe(III) chloride in an aqueous system[67]. Afterward, the precursor was functionalized with the sodium salt of polyacrylic acid (PAA) so that the subsequent silica shell synthesis on the surface of the akaganeite nanorods succeeds without the formation of agglomerates or silica NPs[68–71]. The use of silica-coated akaganeite particles in the subsequent precipitation and ion exchange reaction with cobalt and ferric chloride salts enables a size-tunable synthesis of cobalt ferrite particles in dependence on the silica shell thickness. Using a one-step hydrothermal reaction, these slightly anisotropic NPs will be arranged in nanochains by thin decondensed silica shells around the ordered system. This would also enable the subsequent functionalization of the silica surface, enabling frequently used stabilization in biological media or functionalization with linkers, spacers, and fluorescent substances for magnetic imaging applications[12,72]. Furthermore, the arrangement in long chains results in higher shape anisotropy of the particle system and enables the enhancement of hyperthermia efficiency for cancer treatments due to the higher thermal fluctuations through viscous drag[26].

## Results

In Fig. 1a, the schematic synthesis protocol is shown for obtaining nanorod precursors with a defined silica shell, which are then converted into cobalt-doped ferrite nanoparticles using a hydrothermal step. In the first section, the synthesis of the silica-coated nanorod precursor *via* PAA modification and subsequent Stöber process is illustrated. The silica shell is thoroughly characterized to clarify its impact on the subsequent synthesis of cobalt ferrite. In the following section, the influence of reaction temperature of the hydrothermal step, the filling volume of the reactor, and silica thickness of the precursor on the morphology, particle size, and cobalt content of the magnetic particles is investigated. This is followed by a discussion of the proposed formation mechanism depending on the precursors used. The magnetic characterization, including saturation magnetization, squareness ratio, coercivity, and static susceptibility, is presented in the next part. As the magnetic nanoparticles predominantly form chain-like nanostructures, this study was conducted in aqueous media and in 80% glycerol/water mixtures to investigate the effect of viscosity on potential chain formation, depending on the magnetic field strength. A detailed investigation follows in the subsequent section, as illustrated in Fig. 1b, where the visualization of chain formation and cluster size of the structures as a function of the applied magnetic field strength was achieved through analysis of small-angle X-ray scattering (SAXS) data, depending on the field direction (perpendicular, parallel, and overall). The results obtained regarding the size, size distribution, anisotropy, and chain formation of the synthesized magnetic particles were critically evaluated in comparison to the current state-of-the-art literature, considering their potential biomedical applications.

### Synthesis of modified precursors with different silica shell thicknesses
**Synthesis of the akaganeite nanorod precursors.** Initially, a standard protocol by Hinrichs et al. was used to synthesize well-defined cigar-shaped nanorods of $\beta$-FeOOH (see Supplementary Fig. 1.1a, 1.1b)[67]. These nanoparticles are produced through the hydrolysis of ferric chloride in the presence of $Na_2HPO_4$. As a result, nanorods with an average length of (32 ± 8) nm and a width of (7 ± 2) nm were obtained

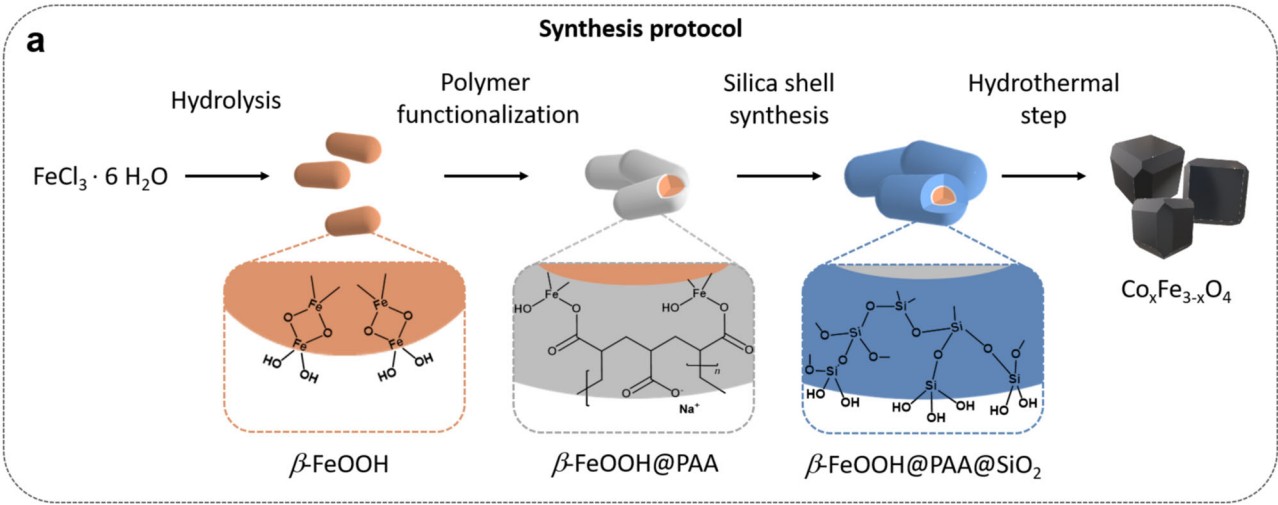

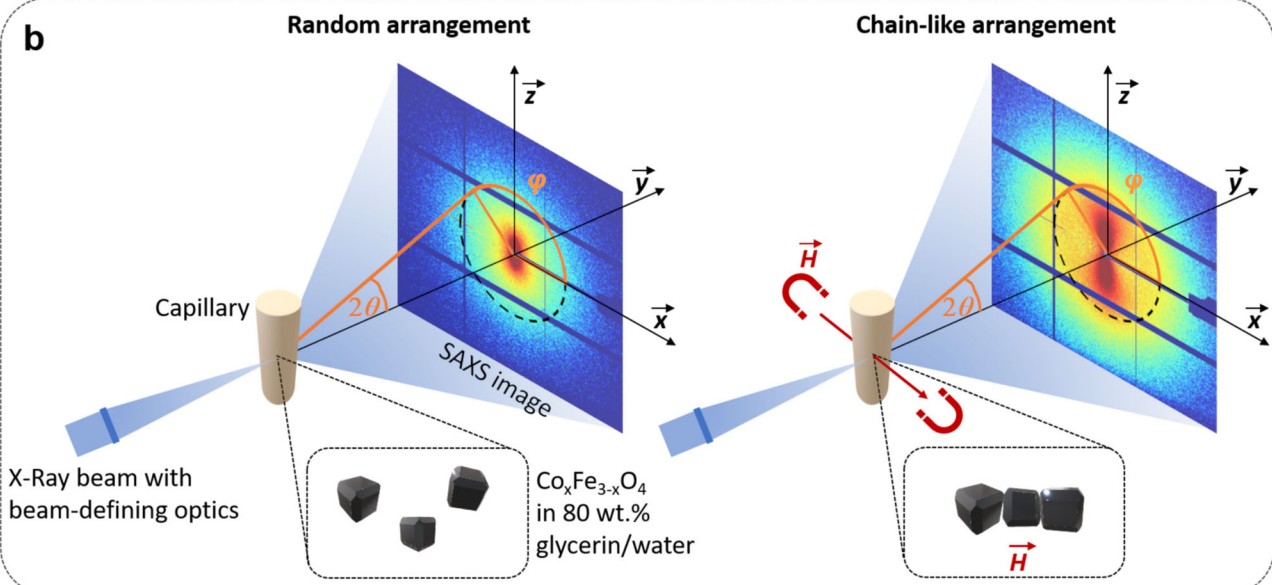

**Fig. 1 | Schematic illustrations of the synthesis and the measurement concept.**
**a** The synthesis protocol for obtaining core-shell nanorods is presented, which involves three steps: the hydrolysis of iron(III) chloride hexahydrate ($FeCl_3 \cdot 6\,H_2O$) to akaganeite ($\beta$-FeOOH), the modification with sodium polyacrylate (PAA), and the deposition of a thin silica layer ($SiO_2$). In a subsequent hydrothermal step, these precursor particles are converted into cobalt-doped ferrite nanoparticles ($Co_xFe_{3-x}O_4$). The boxes visualize the surface coating of the nanoparticles in each step, respectively (orange: akaganeite with hydroxy (–OH) groups, grey: PAA layer with carboxy (–COOH) groups, blue: silica layer with –OH groups). **b** The SAXS images of the random (left) or chain-like (right) particle arrangements in dependence on the magnetic field strength $H$, which is perpendicular to the X-ray beam, are visualized. The chain-like orientation will manifest as an anisotropic pattern, whereas random orientations are generally more isotropic. The patterns were analyzed over all azimuthal angles $\varphi$, perpendicular, and parallel to the field to assess the extent of the anisotropy and to examine the chain-like structures in terms of width and length.

(see Supplementary Fig. 1.1c, 1.1d). The synthesized akaganeite crystal structure is based on double chains that form $2 \times 2$ (and less $1 \times 2$) channels stabilized by typically chloride ions within these tunnel structures. The double chains consist of $Fe(O, OH)_6$ octahedra, which are edge-shared and connected *via* oxo-bridges at the corners. The structure is oriented along the [001] direction, promoting the formation of rod-like structures, schematically illustrated in Supplementary Fig. 1.1e[68,73–75]. The structure-directed role of the chloride ions at the cluster-based akaganeite formation process is well established[76]. This is composed of the thermodynamic driving force to remove chloride ions during dehydration and cluster aggregation of oxyhydroxide clusters, as well as the kinetic hindrance of this process due to the rigidity of the formed subunits with channels[77]. Furthermore, the presence of 4.3 to 6.0 wt.% chloride ions within the tunnel structure (Supplementary Fig. 1.2) prevents the formation of the compositionally comparable goethite at low pH, which is a comparable concentration range as given in the literature[78]. Due to the

pH, the temperature, the anion´s presence, and the requirement for $Na_2HPO_4$ (as discussed in detail in Supplementary Note 1.1), the phase transformation to hematite is prevented as the phosphate ions bind preferentially on (hk0) faces, simultaneously promoting the anisotropic growth along the c-axis[74,79–81]. The diffraction pattern matches the $2\theta$ angles and intensities with reference to JCPDS PDF No. 00-034-1266 in good agreement (see Supplementary Fig. 1.3). The observed diffraction reflexes at $2\theta$ values of 11.8°, 16.8°, 26.7°, 34.0°, 35.2°, 39.3°, 46.5°, and 55.9° correspond to (110), (200), (310), (400), (211), (301), (411), and (521) planes which align with the tetragonal structure (room group I4/m) of akaganeite (see Supplementary Table 1.1). Through the analysis of reflection intensity ratios (see Supplementary Table 1.1) in combination with High-Resolution (HR) TEM analysis (Fig. 2a, Supplementary Fig. 1.4), insights into the crystal orientation concerning the preferred exposed facets of the akaganeite nanorods can be obtained, which is discussed in detail in Supplementary Note 1.2.

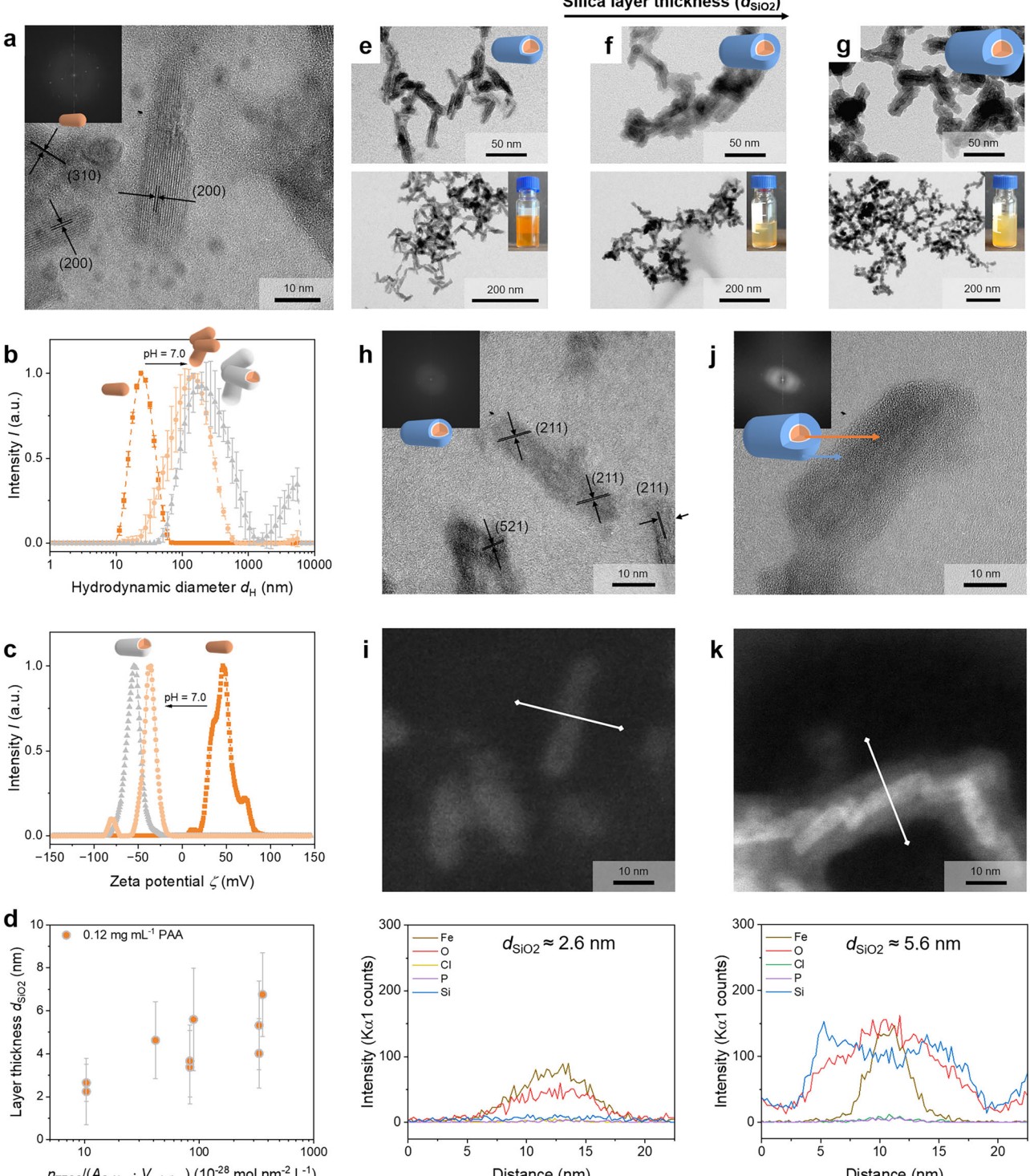

**Fig. 2 | Modified precursors with different silica shell thicknesses. a** HR-TEM image of an akaganeite nanorod with crystal planes analyzed by Fast Fourier Transformation (inset). **b** The hydrodynamic size of the particle system increases with the polymer modification (@pH = 7.0, 0.12 mg mL$^{-1}$, grey) of the akaganeite rods (initial @pH = 2.5, orange). In reference, an akaganeite solution at pH 7 (bright orange) is shown. The hydrodynamic radius was measured thrice for each sample, and the Z-average was calculated. **c** The change of the surface charge in aqueous particle solutions from positively charged iron oxide hydroxide (@pH = 2.5) to negatively charged acrylic acid end groups (@pH = 7.0, 0.12 mg mL$^{-1}$) is indicated by the change of the zeta potential. **d** The silica thicknesses $d_{SiO2}$ are depicted as a function of the molar amount of tetraethoxysilane per surface area and per volume of the reaction solution, exemplified for the aka@0.12PAA precursors. TEM images in different magnetizations and corresponding photos (inlet) of akaganeite@0.12PAA@SiO$_2$ particle solutions with **e** $d_{SiO2}$ = 2.2 nm, **f** $d_{SiO2}$ = 4.7 nm, and **g** $d_{SiO2}$ = 6.8 nm are shown, exemplarily. The precursors with silica shells with **h** $d_{SiO2}$ < 4 nm and **j** $d_{SiO2}$ > 4 nm do not show significant differences in crystallinity, as both cases are assumed to be amorphous, as indicated in the HR-TEM images and by the shape of the line profile scans. Despite the silica shells of **i** $d_{SiO2}$ < 4 nm and **k** $d_{SiO2}$ > 4 nm, the presence of akaganeite is detectable, as confirmed by nanorod areas (top) and corresponding EDX line-scan profiles (bottom).

The hydrodynamic diameter $d_H$ of the akaganeite nanorods is measured to be $(23.0 \pm 0.2)$ nm with a dispersity index (PDI) of 0.15, indicating a monodisperse suspension (Fig. 2b). An aqueous suspension of akaganeite nanorods exhibit a positive surface charge at pH 2.5 (the conditions utilized in the reaction), characterized by a zeta potential $\zeta$ of $(46.8 \pm 2.3)$ mV, which arises from the presence of protonated hydroxyl groups $-OH_2^+$ (orange box in Fig. 1a). In contrast, an akaganeite sample adjusted to a pH of 7.0 using 0.1 M sodium hydroxide solution (the pH value of the subsequent Stöber process) showed an increase in the average hydrodynamic radius to $(105 \pm 1)$ nm. This increase may be attributed to enhanced particle agglomeration resulting from reduced stability, as the particles now exhibit a negatively charged surface with $\zeta = (-39.0 \pm 0.6)$ mV (Fig. 2c), resulting from $-FeO^-$ groups on the surface. Although a $\zeta$ value of $\pm 30$ mV is considered a moderately stable suspension, the size of the blood cells measured by Riddick et al. on the micron scale of this historical value should not be overlooked, where only static interactions are assumed[82]. For particles with a smaller diameter, the attractive Van der Waals forces become more prominent, necessitating higher surface charges to maintain stability[83]. It should also be noted that positive, neutral, and negative groups can simultaneously occur around the isoelectric point, which can lead to particle agglomeration[84].

## Modification of the akaganeite precursors with sodium polyacrylate.

The subsequent surface modification of akaganeite was carried out using sodium polyacrylate (PAA, $M_w = 2100$ g mol$^{-1}$) (see Supplementary Section 2). Considering a cylindrical geometry, the bare nanoparticles exhibit surface areas of approximately 750 nm$^2$ to 800 nm$^2$ per particle batch. Three concentrations of PAA (0.40 mg mL$^{-1}$, 0.12 mg mL$^{-1}$, and 0.04 mg mL$^{-1}$) were tested, referring to the surface area of the particle fractions of $2.0 \cdot 10^{-28}$ mol nm$^{-2}$, $1.9 \cdot 10^{-29}$ mol nm$^{-2}$, or $5.2 \cdot 10^{-30}$ mol nm$^{-2}$, respectively (see Supplementary Fig. 2.1). The successful modification can be attributed to the changes in the surface coating and the hydrodynamic diameter. In contrast to bare akaganeite, the further increase in the hydrodynamic diameter to $(224 \pm 3)$ nm illustrates the successful modification of akaganeite with sodium polyacrylate (PAA, pH 7.0) (Fig. 2b). This is further evidenced by the negative zeta potential of $(-56.3 \pm 1.9)$ mV (Fig. 2c). The p$K_a$ value of polyacrylic acid[85] around 4.5 indicates the deprotonated carboxylic acid end groups of the polymer being present at the surface of the particles at pH 7.0. If steric effects are introduced through a 'protective' shell, such as the anionic polymer PAA with the general formula $[-CH_2CH(COONa)-]$, lower surface charges may be adequate, because particle interactions are hindered by the steric barrier (grey box in Fig. 1a). This may also indicate that the fraction with $d_H \approx 220$ nm corresponds to individually PAA-stabilized particles. However, due to the significantly increased agglomeration observed during TEM preparation in comparison to bare akaganeite, this explanation is rather unlikely. A second size fraction around 2800 nm shows the presence of large agglomerates due to the functionalization with PAA. As a result, the PDI increases to 0.48 for this particle sample, indicating a polydisperse suspension or moderate aggregation, which can also result from the higher pH due to the reaction conditions (in comparison: PDI = 0.31 for akaganeite@pH7.0). Furthermore, Fourier Transform Infrared (FTIR) spectroscopy demonstrated successful surface modification by measuring the characteristic C=O carboxylic acid (1760–1690 cm$^{-1}$) and C–O (1320–1210 cm$^{-1}$) stretching vibrations, as well as the increasing O–H stretching (3780 – 2750 cm$^{-1}$) vibrations (see Supplementary Note 2.1, Supplementary Fig. 2.2a) while the Fe–O bending modes remain unchanged[86–88]. However, it should be noted that all fractions retain their specific akaganeite reflexes, and the crystalline phase is unaltered (see Supplementary Note 2.2, Supplementary Figs. 2.2b, 2.2c).

## Silica functionalization of the PAA-modified nanorod precursors.

In the subsequent step, the silica functionalization was conducted based on a Stöber sol-gel process (see Supplementary Section 3.1). Ammonia was added as a basic catalyst for the PAA condensation and to generate $NH_4^+$ counterion charges at the unbound carboxyl group of the PAA. The base-catalyzed hydrolysis forms silanol groups utilizing the silica precursor tetraethoxysilane (TEOS) near the $NH_4^+$ charges. During the condensation reaction, siloxane bridges are formed, resulting in the production of a silica network (blue box in Fig. 1a) around the PAA-modified nanorods. The silica shell formation controlled by PAA with $NH_4^+$ functional groups was also investigated in detail by Nakashima et al.[89,90] Furthermore, previous research shows that the use of polar solvents leads to a smaller size distribution of the silica shells or silica particles[91,92]. Stöber already demonstrated that this polarity effect results in the smallest particles and size distributions for methanol[93]. However, this solvent was excluded due to its toxicity. A mixture of water and ethanol was used instead, leading to effectively encapsulated nanorods stabilized with PAA in ethanol, as shown in Fig. 2e–g for three different silica shell thicknesses, which would agglomerate intensely in ethanol without PAA modification. The calculation of the silica layer thickness was performed by analyzing TEM images using Supplementary Equation 3.1, assuming that the diameter of the embedded akaganeite rods corresponds to the mean value obtained from Supplementary Fig. 1.1 (see Supplementary Note 3.1, Supplementary Fig. 3.1). The formation of silica shells with a thickness of approximately 2–7 nm, depending on the TEOS amount per surface and per mass concentration, is exemplarily depicted for the particle samples aka@0.12PAA in Fig. 2d. Due to the more intense particle agglomeration of akaganeite at high PAA concentrations because of salting out effects of the polyelectrolytes, as shown in Supplementary Fig. 2.1, the fractions with 0.40 mg mL$^{-1}$ PAA show larger silica-functionalized particle agglomerates with high deviations in the silica thicknesses (see Supplementary Figs. 3.2a, 3.2b). However, the silica layer thickness can be well adjusted using a smaller amount of PAA, represented by aka@0.12PAA or aka@0.04PAA precursors (see Supplementary Figs. 3.2c–3.2o). The volume of ethanol and the amount of TEOS are critical parameters for this process. An excessive quantity of TEOS results in significant aggregation of the nanorods, which is accompanied by the precipitation of particles. This phenomenon is also observed with insufficient dilution of the synthesis solutions, as the particles are not sufficiently spatially separated, and the silica layers are formed around a bulk of nanorods. A concentration of 2.6 mg nanorods per mL ethanol yields reproducible and visually non-agglomerated particle suspensions; no precipitation was observed in this case (see photos in Fig. 2e–g). Furthermore, when utilizing identical ratios of the amount of TEOS to the product of (product of particle surface area and volume of ethanol), the resulting silica thicknesses by using low PAA concentrations are slightly smaller compared to those at medium or high PAA concentrations (see Supplementary Fig. 3.2p, Supplementary Fig. 3.3). This occurrence is hypothesized to be attributed to reduced particle agglomeration at low PAA concentrations, as a higher amount of TEOS is consumed per particle (or surface) than in an agglomerate formed by multiple particles (see Supplementary Note 3.2).

To demonstrate the presence of akaganeite in the core of the silica precursors and to evaluate the porosity of the silica layers, two fractions were analyzed with $d_{SiO_2} < 4$ nm and $d_{SiO_2} > 4$ nm (see Supplementary Note 3.3). In the HR-TEM images, a clear distinction between akaganeite and silica can be observed (Fig. 2h and j, see Supplementary Fig. 3.4). Additionally, Energy Dispersive X-ray spectroscopy (EDX) analysis reveals silicon signals at the surface and edges of the particle, whereas iron is primarily present in the core, especially for the sample with the thicker shell. The transition is clearly visible in the line-scan profiles (Fig. 2k, see Supplementary Fig. 3.5). However, in the sample with the thin silica shell, where the line scans do not appear to be very conclusive (Fig. 2i, see Supplementary Fig. 3.6), a Si/Fe ratio of 0.28 is observed based on the distribution in the sum spectra (see Supplementary Fig. 3.7). This is significantly different from the Si/Fe ratio of 0.09 that was determined in the bare akaganeite sample (see Supplementary Fig. 1.2). In the sample with a thicker shell, the silicon content correspondingly increases, resulting in a Si/Fe ratio of 2.7 (see Supplementary

Fig. 3.7). Chloride atoms, located within the tunnel structures of akaganeite, can also be detected. This confirms that akaganeite remains stable and is neither dissolved nor transformed into another phase during the PAA functionalization and the Stöber process. Additionally, the HR-TEM analysis enables a cautious interpretation of the porosity of the silica shells, which lack crystal planes and are therefore more likely to exhibit an amorphous structure. Additionally, the line scan profiles do not show a single mountain profile of Si and O. Therefore, they cannot be assigned to a crystalline phase. Instead, they exhibit an amorphous structure with non-ordered mesopores, as demonstrated in Rades et al. for silica spheres[94]. This is attributed to differences in the local particle density, corresponding to silica with either aligned or non-aligned mesoporous structures[94]. Comparable porosities were also investigated and determined using gas adsorption measurements in combination with Brunauer-Emmett-Teller analysis for similarly synthesized silica shells *via* the Stöber process (see Supplementary Note 3.4). Such a structure could dissolve under alkaline conditions, with the dissolution time depending on the thickness of the silica layer[95]. Given that the reproductive formulations for the silica functionalization were conducted using 110 mg of nanorods and 100 mg or 55 mg per batch were required for the hydrothermal synthesis, the analysis of the other silica fractions used for hyperthermal steps was restricted to TEM.

### Size-dependent synthesis and arrangement of cobalt ferrite chain-like structures

**Overview**. The synthesized nanoparticles, resulting from the subsequent conversion of different precursors, were investigated in terms of their morphology, size, and elemental composition. The calculated values for size obtained by TEM and X-ray diffraction (XRD, calculated *via* the Debye-Scherrer equation and the Williamson-Hall plot, see Supplementary Note 4.1) and cobalt content obtained by flame atomic absorption spectroscopy (F-AAS, see Supplementary Note 4.2) are summarized in the Supplementary Table S4.1. The dispersity is given in Supplementary Table S4.2. Reference samples were prepared using either bare akaganeite or aka@PAA precursors, resulting in cobalt-doped particles designated as CF-A or CF-P. The designations are combined with a series number, which is explained by the reactor parameters used, where "1" describes a conversion at a maximum temperature of 160 °C and filling volume of 100%, "2" a conversion at 190 °C and filling volume of 100%, and "3" a conversion at 190 °C and filling volume of 55%. Cobalt-doped particles obtained from the reaction of aka@PAA@SiO$_2$ precursors are designated as CF-S. For series 1 and 2, the silica layer is 2.5 nm. In series 3, the silica thickness of the different precursors varies by approximately 2 – 7 nm.

Overall, the resulting magnetic particles are slightly anisotropic, exhibiting cubic or quasi-spherical shapes in the 2D projection (see Supplementary Figs. 5.1, 5.2, and 6.1 for series 1, 2, and 3, respectively). The observed diffraction reflections at 18.2°, 30.0°, 35.5°, 43.0°, 56.9°, and 62.2° at the diffraction patterns (see Supplementary Figs. 5.3a, 5.3b and 6.2 for series 1, 2, and 3, respectively) correspond to the (111), (220), (311), (400), (511), and (440) planes, respectively, which closely match the cubic spinel structure of cobalt ferrite reference (JCPDS No. 00-003-0864) for all approaches. The elemental analysis results indicate the formation of cobalt ferrite with a cobalt-to-iron ratio of 0.16 to 0.20, which means a stoichiometric cobalt coefficient of approximately 0.40 to 0.51, respectively (see Supplementary Table 4.1). The effect of different temperatures and silica layer thicknesses will be discussed in the following two sections.

**Effect of the hydrothermal temperature**. TEM images of the particles are shown in Fig. 3a, b in dependence on the used precursor. As the shape of the particle fractions differs slightly by using akaganeite only or akaganeite with silica shells, the particle initiation and growth mechanism must change, which will be discussed in detail in the Section '*Explanation of the Proposed Mechanism.*' Comparison of the magnetic particle shapes synthesized with aka and aka@PAA shows a more angular morphology at lower temperatures. In contrast, at higher temperatures, the particles

tend to become more rounded. However, it is particularly well known that Ostwald ripening predominantly occurs at the edges and corners of cubic and octahedral particles. Consequently, the morphology changes to significantly truncated particles with higher temperatures, as can be observed in the TEM images. Additionally, the overall particle distribution changes (see Supplementary Figs. 5.1, 5.2) to slightly structured nanochain orders using akaganeite-core/silica-shell particle systems.

A comparison of the particle size values determined from TEM images and the calculated crystallite size $d_{DS}$ observed from the diffraction patterns *via* the Debye-Scherrer equation, in dependence on the cobalt-to-iron ratio, is illustrated in Fig. 3c and d. The dispersity index from the TEM data, at approximately 0.10, is comparable across all experiments, suggesting no explicit dependency of the reaction temperature on the size distribution. Observations from both temperature approaches indicate that the diameter determined from the diffraction pattern is notably smaller than the diameter from the TEM images. The particle size distribution has been reproducible.

The diffraction patterns are presented in Supplementary Fig. 5.3, showing that the most intense peaks closely match those of the reference sample (JCPDS No. 00-003-0864), confirming the formation of cobalt ferrite and the experiment's reproducibility. The exact cobalt content obtained *via* F-AAS ranges indicates the formation of cobalt-doped particles with $x < 1$, showing minor discrepancies from the initial trial but within the error range of ±0.01, and increases with the particle size. While the blank samples exhibit the lowest cobalt content in both temperature profiles, the cobalt content in the repeated experiments shows slight deviations.

The mean values from the replicate experiments (see Supplementary Table 4.1) using silica-functionalized precursors at 160 °C and 190 °C confirm an increase in cobalt content and diameter with higher temperatures. The mean values and deviations for the Co/Fe ratios and the diameter of both samples are $\kappa = 0.188 \pm 0.008$ and $0.188 \pm 0.006$, as well as $d_{TEM} = (35.1 \pm 8.2)$ nm and $(40.9 \pm 6.1)$ nm and $d_{WH} = (37.6 \pm 3.0)$ nm and $(41.9 \pm 3.8)$ nm for 160 °C and 190 °C, respectively (see Supplementary Fig. 5.4 for WH plots). Comparable cobalt coefficients ranging from 0.46 to 0.49 are identified at both temperatures. However, the distribution of nanoparticle sizes at both temperatures is quite similar, making it difficult to draw significant conclusions. Reproducibility appears to improve at high temperatures, with cobalt contents and diameters aligning more closely. The temperature difference is a crucial factor, particularly when using a filling volume of 100%, as higher temperatures enable higher heating rates (see Supplementary Fig. 5.5), allowing for crystalline cobalt ferrite formation 15 minutes earlier[96,97]. This enables the synthesis of thicker cobalt ferrite shells or an improved ion exchange with higher reaction temperatures, leading to slightly larger diameters and higher cobalt contents. The impact of this temperature difference is more prominent with a filling volume of 55%, where a temperature above 80 °C is attained after 15 minutes of heating. This explains the use of higher temperatures with a smaller filling volume for series 3.

The discussion of the temperature effect will be based on the mean values of both trials. The particle fractions produced using silica precursors suggest that the low temperature (160 °C) favor the formation of smaller sizes. This could be related to the dissolution of silica structures under alkaline conditions, with the dissolution rate depending on the thickness of the silica layer[95]. The amount of dissolved silica increases at high temperature (190 °C), due to an enhanced dissolution rate and higher saturation concentration of silica during the reaction in water[95]. At high temperature, coupled with increased amounts of dissolved silica, there is less spatial separation of nuclei during the particle growth process, allowing for the formation of larger particle diameters. Conversely, increased salinity of the solution enhances the stability of the silica shells, which may preferentially occur at low temperature and the associated lower kinetic energy in the system. This would result in a higher proportion of ionic compounds, such as unreacted Fe(III), Fe(II), and Co(II) ions, remaining in the aqueous solution with a lower activity coefficient of water over a longer period of time. The silica layers would still be present in the solution and support the

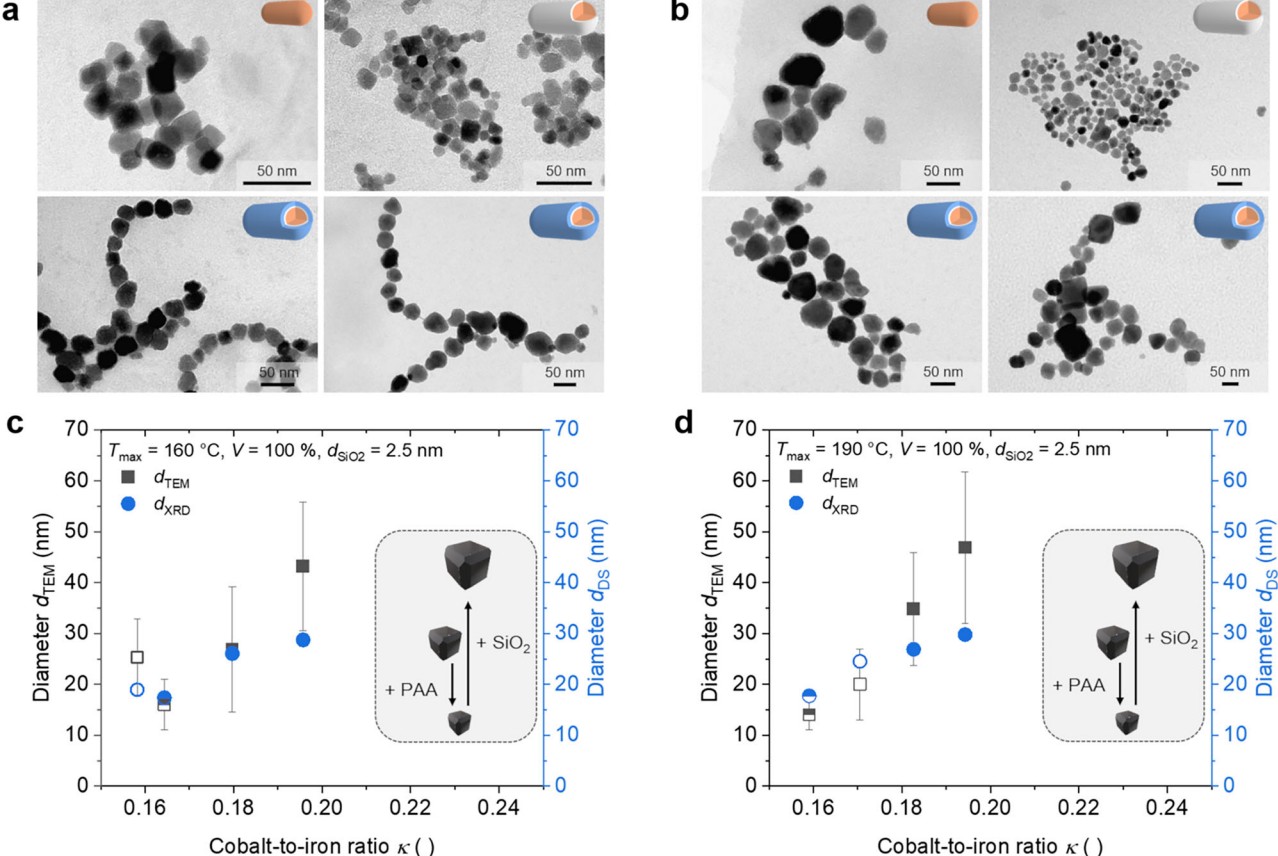

**Fig. 3 | Comparison of the shapes, sizes, and cobalt contents of the cobalt-doped ferrite samples by varying the maximum reaction temperature.** The bare akaganeite (hollow), aka@0.40PAA (half-filled), or aka@0.40PAA@2.5SiO₂ (filled, with a shell thickness of 2.5 nm) precursors were converted in a hydrothermal step with a filling volume of 100%. The conversion of aka@0.40PAA@2.5SiO₂ precursors was performed twice. **a** At a maximum reaction temperature of 160 °C, the shape changed from cubic to truncated cubic shapes. **b** By increasing the reaction temperature to 190 °C, the shape is more truncated. In comparison to the use of bare akaganeite, the size and cobalt-to-iron ratio of the cobalt-doped particles decrease when using PAA-modified precursors, and increase when a silica shell is added around the precursors for both setups, performing the reaction at either **c** 160 °C or **d** 190 °C.

formation of magnetic particles close to these silica shells, thereby favoring the development of more chain-like structures.

**Effect of the silica shell thickness.** The TEM images of the cobalt ferrite samples synthesized with varying shell thicknesses (series 3, see Supplementary Section 6) are displayed in Fig. 4a, and overviews of different magnifications are shown in Supplementary Fig. 6.1. As indicated in the HR-TEM images, it can be assumed that the particles exhibit octahedral morphologies as presented for the sample CF-S3-8 (see Supplementary Fig. 6.2). Specifically, the cobalt ferrite diameters notably increase with the silica thickness of the utilized precursors, except for the bare precursor (Fig. 4b). This deviation could be attributed to the additive (PAA polymer), which is used for the samples with a silica shell. The literature suggests that additives can result in smaller crystallite sizes[98–101]. Therefore, we also synthesized a blank sample with a precursor without a silica shell but containing PAA instead. This precursor results in the smallest diameters for these NPs. Further support of this trend is observed by the crystallite size determined by the Gaussian fit of the (311) reflex of the diffraction patterns (see Supplementary Fig. 6.3) as well as by the Williamson-Hall plot for all reflexes (see Supplementary Fig. 6.4). Furthermore, the cobalt content of the samples increased with the silica layer thickness and the nanoparticles´ size, respectively (Fig. 4c). However, the sample using aka@PAA showcased another observation as the diameter shrinks and the cobalt content increases. Compared to this blank sample, the diameters and cobalt contents increase with increasing shell thickness. A significant enhancement is observed at a silicon thickness of 4 nm.

Furthermore, a shift in the $2\theta$ angles of the XRD patterns can be observed (see Supplementary Fig. 6.3b). When plotting the most intense peaks at the (311) reflex of the cubic inverse spinel (Fig. 4c), a slight shift of approximately 0.1° to smaller angles is initially observed with the addition of PAA. This continues with the use of precursors with a silica thickness of up to 4 nm up to 35.44°. As soon as the layer thickness exceeds 4 nm, a jump back to 35.57° can be recognized, and it starts shifting again to 35.44° with increasing silica thickness. If cobalt ferrite (ICPDS No. 00-022-1086, $2\theta_{(311)}$ at 35.44°) is referenced, agreement with this can be found for almost all samples. Magnetite shows a maximum of $2\theta_{(311)}$ at 35.42° (ICPDS No. 00-019-0629). The deviating sample at 3.4 nm ($2\theta_{(311)} = 35.34°$) can be explained by the lower Co/Fe ratio of 0.16, which would explain a shift in the reflexes to more magnetite-like ferrites. Because X-rays primarily resolve the surface of the crystal lattice structure, cobalt-rich surfaces will exhibit a shift of the (311) reflection to a $2\theta$ value of 35.44°. To confirm the Co/Fe ratio, line scans and EDX mappings were additionally performed on three samples, namely CF-A3 (Fig. 4d), CF-P3–1 (Fig. 4e), and CF–S3–8 (Fig. 4f). The CF samples show the same trend as in the F-AAS measurements, namely increasing Co/Fe ratio with PAA addition and further enhancement with increasing particle diameter. However, it can be observed that the absolute $\kappa$ values of 0.205, 0.213, and 0.222 observed from the map sum spectra are higher than those determined by AAS. This discrepancy is due to the partial overlap of the Co and Fe Kα1 lines, given inherent uncertainties in the intensity values. Additionally, differences in the average values observed through the line sum spectra and the map sum spectra can be identified

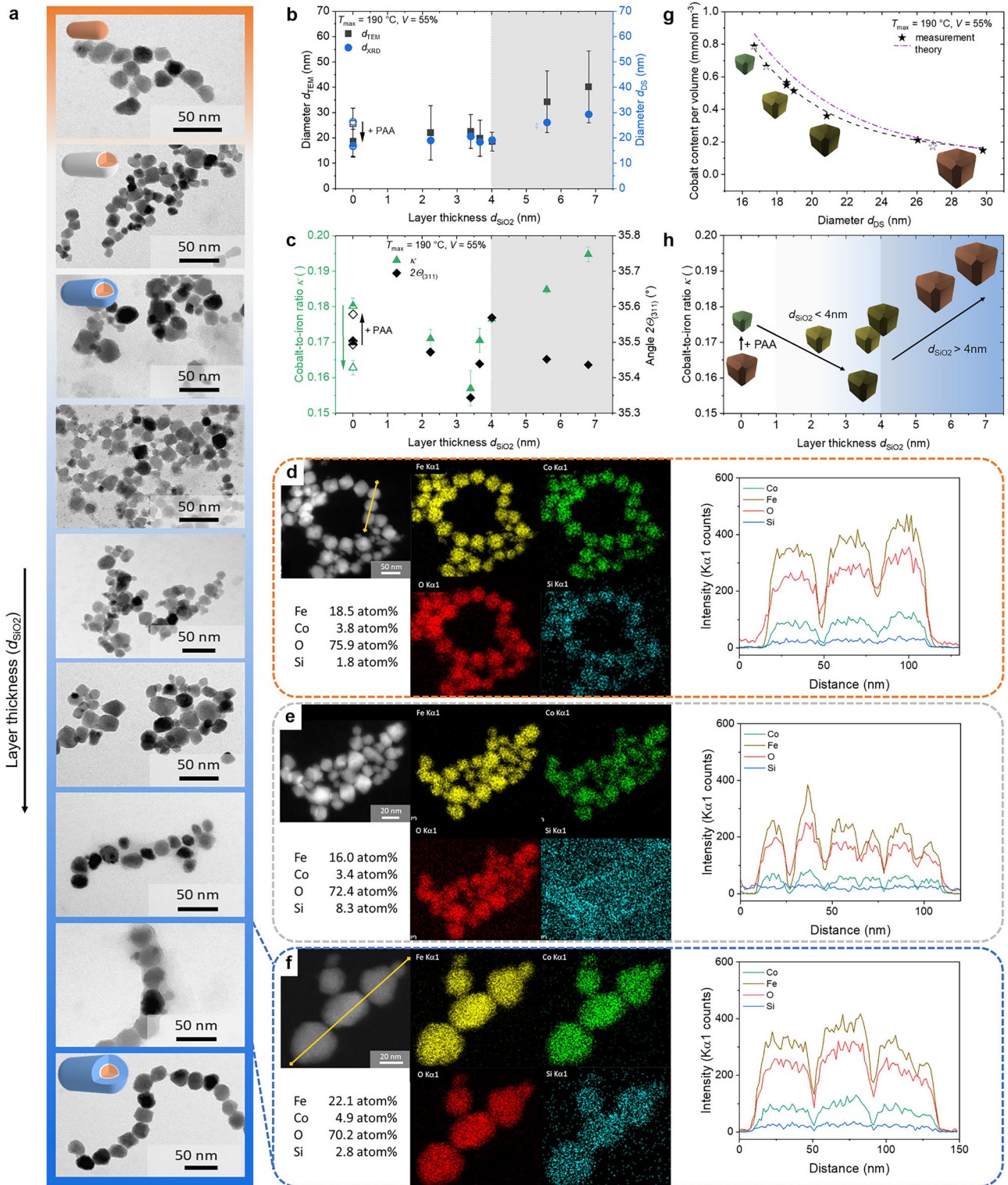

**Fig. 4 | Comparison of the shapes, sizes, and cobalt contents of the cobalt-doped ferrites by changing the thickness of the silica shell of the precursors.** The akaganeite (hollow, orange), aka@0.12PAA (half-filled, grey), or aka@0.12PAA@SiO$_2$ (filled, blue) precursor with a shell thickness varying between 2.2 nm and 6.8 nm was converted in a hydrothermal step with a filling volume of 55% at a temperature of 190 °C. **a** TEM images indicate a change in the shape from cubic to truncated cubic particles by increasing the silica shell of the precursors. **b** The morphological and crystallite diameter, as well as **c** the cobalt-to-iron ratio of the resulting cobalt-doped particles, increase with the silica shell thickness. EDX mapping and corresponding intensity line scan profiles of the cobalt-doped particles synthesized with **d** akaganeite, **e** aka@0.12PAA, or **f** aka@0.12PAA@5.6SiO$_2$ confirm the presence of Co (green), Fe (yellow), O (red), and

Si (turquoise). However, the distribution includes only the intensities of Si, Co, Fe, and O, which were normalized to 100 atom%. Therefore, the absolute values should be lower when considering all elements, as other components are not included in this normalization. **g** The formation of MNPs from bare akaganeite indicates the presence of a magnetite-like phase (brown particle). Smaller MNPs with a high cobalt content, indicated by a green color, can be synthesized by adding a polymer additive. The diameter and cobalt content per unit volume can be adjusted by converting a precursor system with a silica shell. As the cobalt content in the suspension increases, along with the thickness of the silica shell and the diameter of the particles, the mechanism of formation of the magnetite core changes. **h** Fig. 4c is depicted in dependence on the size and cobalt content per volume, summarizing the data from F-AAS, XRD, and TEM.

(see Supplementary Figs. 6.5–6.10). The mapping, which analyzes above a larger number of particles, yields a value that approaches the AAS value (according to the significantly larger particle number). Furthermore, the atom percent distribution of the line scans enables an interpretation of the silicon presence. Interestingly, in the case of CF-S3-8, the silicon intensity at the particle edges increases, which can be attributed to a local increase in silicon concentration (see Supplementary Fig. 6.9). This indicates the presence of a thin silica shell surrounding the particles, which stabilizes them within the chains or at least influences their formation mechanism (see the following section).

In particular, it should be noted that all trends regarding the size, the cobalt amount, and the $2\theta$ angle at the maximum of the (311) reflex show a jump in the data trends as soon as a silica thickness of around 4 nm is exceeded. To understand the interplay of the parameters, the cobalt content was calculated over the entire particle volume, assuming a simplified cubic particle shape (Fig. 4g). The theoretical value of the iron-to-cobalt content of 0.20 is derived from the amount of Co(II) ions used concerning the maximum usable Fe(III) and Fe(II) content (see Supplementary Note 7.1). If it is further assumed that the particle shape is approximately cubic, the cobalt content per cubic volume of a particle decreases in relation to the increasing silica thickness or size of the particle. It is evident that the theoretical value of cobalt content per particle volume is nearly achieved, with only a minor deviation observed for smaller diameters. This could be explained by minor deviations in the determination of weight percentages, which could result in slight discrepancies in the mass concentration of 10 mg MNP per g solution in the AAS measurements. Additionally, it is conceivable that small amounts of Co(II) ions in solution may not be incorporated due to an ion exchange at high temperatures, which could lead to a more significant deviation from the theoretical value with a higher number of smaller particles. A clear inverse relationship exists between particle size and cobalt distribution, with cobalt-rich particles depicted in green and iron-rich phases in brown. Considering only the distribution over the total volume, it is evident that, at the same amount of cobalt in the reaction solution but with different resulting particle sizes, the cobalt content should decrease with increasing particle size. Additional analysis of the (311) reflections provides indications of the iron- or cobalt-rich surfaces of the particles, which differ from the simplified depiction in Fig. 4g (see comparison in Fig. 4c and discussion above). In Fig. 4h, the particle size and cobalt distribution over the particle volume are illustrated, along with the cobalt-to-iron ratio, which summarizes the trends discussed above. The effects on particle size, cobalt content, and arrangement due to the addition of PAA and increasing silica thickness are shown in Supplementary Fig. 7.1.

**Explanation of the proposed mechanism.** We hypothesize that the combined effects of the size, cobalt content, and surface occur because different formation mechanisms for magnetite in the initiation phase are feasible within the present reaction solution, along with a varying dissolution process of the akaganeite rods due to different silica shell thicknesses (Fig. 5). The initiation phase proceeds at room temperature within the first seconds of NaOH addition, in which nuclei spheres of magnetite occur. The spherical shape is present because the surface energy and surface area should be minimized, as surface tension is the dominant parameter in the initiation process[102]. Magnetite can be formed through a basic coprecipitation reaction of Fe(II) and Fe(III) hydroxides (Massart reaction), depicted *via* reaction 1. Moreover, akaganeite can be converted to magnetite with Fe(II) hydroxide, provided the Fe(III) concentration in the solution remains above the solubility limit of akaganeite of 10 mmol L$^{-1}$ (reaction 2)[66]. Both mechanisms take place at room temperature. In the absence of an additional silica layer around the akaganeite nanorods, these transformations are scarcely affected, and magnetite seeds are generated upon the addition of sodium hydroxide. Subsequently, particle growth proceeds *via* diffusion of metal ions toward the nuclei. This process induces a decrease in the concentration of the metal ions within the solution, which exerts a significant influence on the solid precursor, particularly in the case of Fe(III). The precursor dissolves

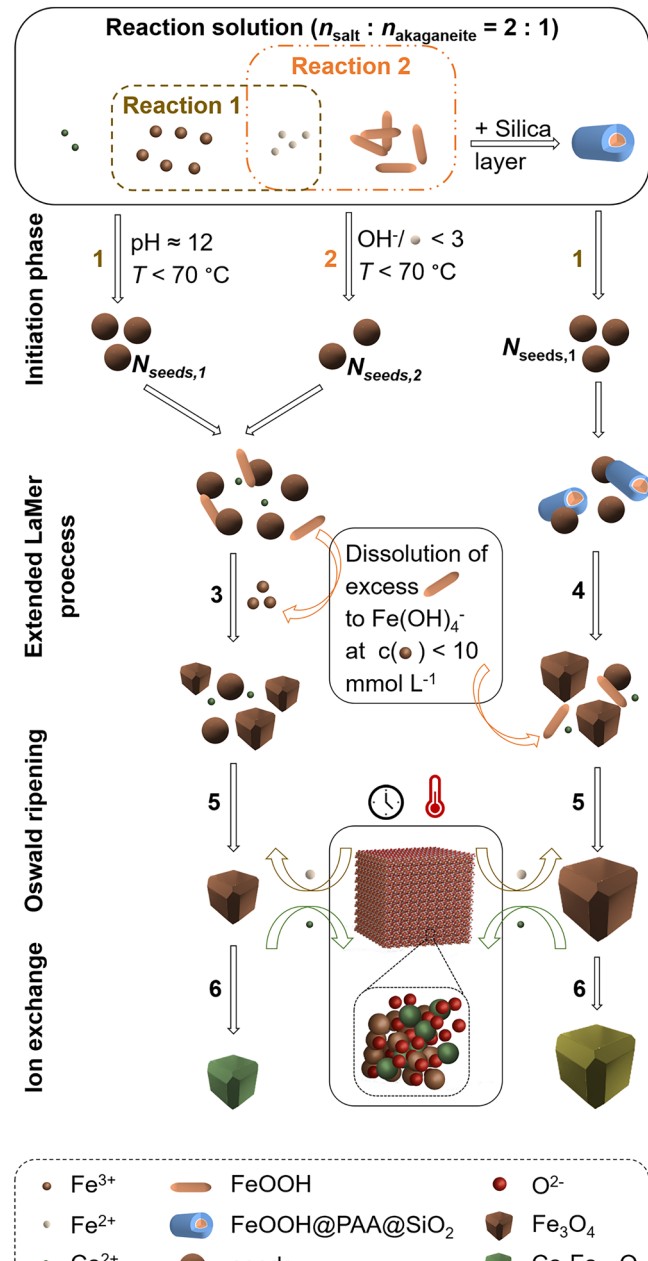

**Fig. 5 | Scheme of the proposed mechanism.** Reaction (**1**) depicts the Massart reaction from Fe$^{2+}$ and Fe$^{3+}$ hydroxide to obtain a number of magnetite seeds $N_{seeds,1}$. The precipitation (**2**) of bare akaganeite with Fe(OH)$_2$ leads to the formation of magnetite seeds $N_{seeds,2}$. The extended LaMer process (**3**) takes place at $c$(Fe$^{2+}$) < 10 mmol L$^{-1}$, where excess akaganeite rods dissolved to Fe(OH)$_4^-$ to ensure a consistent supply with Fe$^{3+}$. Additionally, the Oswald ripening process (**5**) can occur, whereby the smaller particles dissolve to promote the growth of the larger particles. During the hydrothermal step, the ion exchange reaction (**6**) between Fe$^{2+}$ and Co$^{2+}$ results in the synthesis of cobalt-doped particles. If the akaganeite precursor is protected by a silica shell, (**2**) does not occur, and the dissolution of the nanorods (**4**) is significantly delayed. As a result, a decreased number of nuclei and a markedly increased particle growth are presented.

when Fe(III) concentrations fall below 10 mmol L$^{-1}$, provided it has not been entirely precipitated during reaction 2 (see below for discussion on the molar ratio between the precursor and the metal salt). This dissolution establishes an additional Fe(III) reservoir during the growth phase, often referred to as an extended LaMer model (reaction 3)[103,104]. Furthermore, Ostwald ripening, which is driven by the higher surface energy

of smaller particles, facilitates their dissolution and subsequent redeposition onto larger particles (reaction 5)[105,106]. This phenomenon promotes particle size focusing, resulting in a narrowing of the particle size distribution over time. During the entire duration of the hydrothermal step, Co(II) ions can be integrated into the crystal lattice (ion exchange with Fe(II) ions), leading to a uniform distribution of Co(II) throughout the particle volume (reaction 6).

As a result of the left pathway depicted in Fig. 5, particles with an approximate size of 27 nm and an average cobalt content of 0.2 mmol per nm$^3$ are formed by using akaganeite precursors only. As the particles become more monodisperse and smaller (approximately 17 nm) due to the addition of PAA, the surface-to-volume ratio increases significantly, resulting in a relatively intensified cobalt incorporation at the surface. The cobalt content per volume increases to an average value of 0.7 mmol per nm$^3$. Both the decreased size and size deviation value can be attributed to the specific interactions of additives, which primarily serve to inhibit excessive agglomeration of the seeds and particles. By enabling spatial separation, the number of nuclei increases, leading to a decrease in particle size. Based on this mechanistic understanding, the addition of PAA by modification of the akaganeite precursors is expected to significantly reduce the agglomeration of magnetite seeds, with nanorods acting as a source of iron ions during the particle growth process.

Additionally, not only the PAA effect must be considered, but also the silica thickness of the precursors. As the silica thickness increases up to a value of approximately 3.3 nm, the particle size remains relatively constant ( ~ 20 nm). We assume that the transformation of akaganeite is delayed, regarding the dissolution process as well as the precipitation process, leading to a smaller number of nuclei. In order to achieve this decrease, the smaller number of nuclei can grow to larger ones compared to the particles synthesized via the mechanism without a silica shell. This can also allow ion exchange to occur simultaneously during the conversion, leading to a more consistent distribution of the cobalt amount throughout the entire particle volume. This distribution value decreases to a value in the range of 0.6 and 0.4 mmol per nm$^3$, and the $2\theta$ values also decrease to iron-rich surfaces. At a thickness of 3.3 nm, there is a pronounced drop in cobalt content, which we attribute to a potential reversal of the mechanism. The temporally delayed release of the akaganeite particles following the silica "protective" layer's dissolution results in the formation of a more iron-rich layer around the cobalt-doped MNPs. This is accompanied by a significant reduction of the $2\theta$ angle. However, a markedly reduced total cobalt content (determined via AAS) was detected in this sample, which may correlate with an anomalous mass concentration determination of MNPs of 1 wt.%. At a silica thickness of 4 nm, the silica layer surrounding the akaganeite nanoparticles becomes sufficiently thick. Thus, the nanorods are protected from conversion with Fe(II) to magnetite. Consequently, only magnetite nuclei are initially available, which are formed through the Massart reaction (right pathway). The subsequently available akaganeite is then utilized with Fe(II) hydroxide for crystal growth at elevated temperatures, resulting in the formation of significantly larger particles up to 30 nm due to the smaller number of nuclei (reaction 4). Akaganeite could also act as a solid reservoir and be dissolved through a decrease in Fe(III) concentration in the solution, as explained above. However, this process would occur with a delay compared to the pathway on the right. Both effects would exhibit a reduced surface-to-volume ratio due to the higher volume, leading to a pronounced overestimation of the cobalt ions at the surface via XRD compared to the cobalt amount-to-volume ratio. This ratio decreases to around 0.20 mmol per nm$^3$ for the largest particles. As the cobalt-to-iron ratio ($\kappa \leq 0.20$) of the particle fractions decreases with the addition of PAA and increases with the thickness of the silica shell, we can presume a correlation between the size and cobalt content within the crystal structure.

Furthermore, the metal salts and akaganeite precursor amounts are employed in a 2:1 ratio, which should be considered in the mechanistic discussion. Following the addition of NaOH, magnetite is formed from the akaganeite precursor at room temperature. However, the nucleation phase is prolonged instead of the typical rapid initiation and nucleation phases. With

a thicker silica shell, nucleation is delayed, leading to the development of magnetite core particles of different sizes. Cobalt is integrated into the magnetite phase through ion exchange, provided there are no excess Fe(III) or Fe(II) ions in the solution. This process results in creating a mixed ferrite layer on the magnetite surface, as evidenced by the reduced cobalt content compared to stoichiometric co-precipitation experiments. The delayed initiation step also accounts for the increased size distribution in all reactor experiments compared to the blank samples. Moreover, with a higher proportion of Co(II), Fe(II), and Fe(III) ions than necessary for akaganeite consumption, a cobalt ferrite shell can encase the mixed ferrites, leading to a more spherical shape of the mixed ferrites in contrast to the cubic shape observed in the blank samples. Assuming that all akaganeite is converted to magnetite (0.619 mmol of akaganeite with 0.310 mmol of Fe(II)), a mixed ferrite shell composed of magnetite:cobalt ferrite could be formed in ratios ranging from 2:1 to 1:2. Further mechanistic change might be, that the silica shell inhibits the formation of magnetite due to the basic precipitation of akaganeite precursors and Fe(II) ions. The nucleation of magnetite will take place preferentially via the basic coprecipitation reaction between Fe(III)- and Fe(II)-ions. As a result, the akaganeite precursors would be present at higher temperatures in the reaction solution and could be converted to magnetite or cobalt ferrite particles. However, this is unlikely, as we do not observe additional spherical particle species via DLS or TEM. More likely, the akaganeite nanorods dissolve and are now forming the mixed cobalt ferrite shell around the magnetite cores using excess Co(II), Fe(II), and Fe(III) ions. If the conversion of akaganeite is delayed by the silica shell and magnetite cores are formed from the iron salts present in the solution, the magnetite:cobalt ferrite ratio for the magnetic shell would change only marginally.

The cobalt-doped ferrite formation process explains the enhancement of the nanoparticle size and the reduction of the cobalt content per volume with increasing silica shell thickness, as discussed above. Since the chain formation, which is observable via TEM (though with limited conclusiveness), appears to increase with the silica shell of the precursor, it is likely that this layer influences the process. Given that the pH value of 12 is established after the addition of NaOH to the reaction solution, the silica shell should gradually dissolve over time[95]. This also explains the dissolution of the akaganeite particles at a later stage of the reaction than without a silica layer. As the formation of the silica layer is a dynamic, base-catalyzed process, we assume that this occurs concurrently during the synthesis of cobalt ferrite particles. After the reaction, the pH value should be lowered due to the dissolution of silica species and the presence of chloride ions. Consequently, a reformation of the silica shell surrounding the magnetically interacting particles should arise. During the extended LaMer process, aggregation occurs spatially close to the particles, facilitating increased chain formation. If more silica is present, this process could be amplified, leading to enhanced chain formation, as the silica shells promote re-coating by hydrolysis and condensation reaction and magnetic interaction between the simultaneously synthesized particles.

## Magnetic behavior of self-arranged nanochains in dependency on the viscosity

To investigate the magnetic properties of the chain-like structures, vibrational sample magnetometry (VSM) measurements were performed from the 1 wt.% sample dispersions in water (Fig. 6a, see Supplementary Note 8.1, Supplementary Fig. 8.1). These investigations enable the calculation of size-dependent interactions, which are contributed by, e.g., magnetic dipole-dipole interactions and Van der Waals forces. These interactions affect the self-assembly behavior of the nanoparticles[25,107,108].

For increasing magnetic field strengths $H$ up to 1.5 T in water, the hysteresis behavior of a ferrimagnetic material occurs as expected[1]. This indicates that the effect (orientation of the magnetic moments of the particles) lags behind the cause (application of a magnetic field of specific strength). Above these magnetic field strengths, the material's magnetization initially increases spontaneously until saturation is reached (Fig. 6a (inlet): transition from phase II to phase III with increasing $H$). We assume

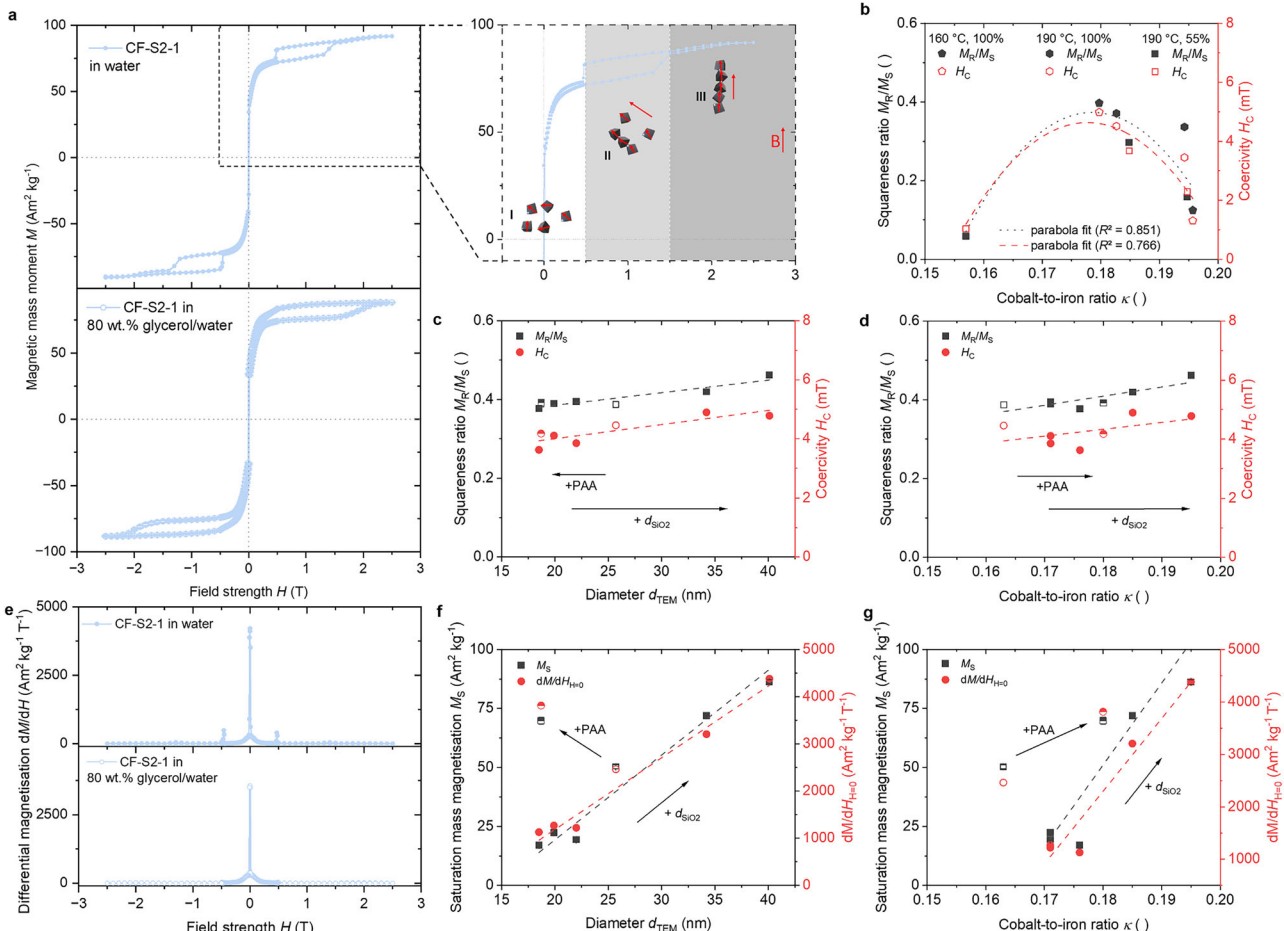

**Fig. 6 | Hysteresis of the magnetic chain-like structures. a** The magnetic behavior comprises stabilizing exchange interactions among the individual particles and the stabilizing effects of the nanochains themselves. The alignment of the nanochains results in an additional drop in magnetization with increasing magnetic field strength (see dashed inlet), exemplified for an aqueous or an 80 wt.% glycerol/water dispersion of sample CF-S2-1 (bright blue), synthesized with the precursor aka@0.40PAA@2.5SiO₂. **b** The squareness ratios and coercivity values in water exhibit the same parabola trend, with these values increasing with the cobalt-to-iron ratio until a maximum is reached at approximately 0.18, synthesized at a temperature of 160 °C and a filling volume of 100%. With increasing **c** diameter and

**d** Co/Fe ratio of the samples dispersed in 80 wt.% glycerol/water, the squareness ratios and coercivities of the CF-S samples (synthesized with different silica thicknesses, filled) are adjustable compared to those of CF-A (synthesized with aka, hollow) and CF-P samples (synthesized with aka@PAA, half-filled). **e** Using the hysteresis loops, the differential magnetization of the exemplary CF-S2-1 curves can be calculated, which shows a maximum at $H = 0$ (static mass susceptibility). The saturation magnetization and the static mass susceptibility of all samples in 80 wt.% glycerol/water are depicted in dependence on the **f** diameter and **g** Co/Fe ratio, recording a rising trend for each.

that with increasing field strength, the individual particles aggregate into ordered nanochains. Microscopically, this behavior is analogous to the Barkhausen jumps of electron spins in a magnet, where individual regions (Weiss domains) collectively reorient their alignment[109–111]. Concerning the present ordered system of the nanochain arrangement, the stabilization of the individual magnetic moments of the particles within this structure occurs in addition to the exchange interaction, resulting in a higher energy requirement for flipping the moments compared to individual nanoparticles. Upon reaching a critical magnetic field strength of approximately 1.5 T, the magnetic moments of the individual particles are collectively realigned in nanochains, forming a head-to-tail arrangement of magnetic dipole moments, resulting in a discontinuous increase in the magnetic moments[32]. Using HR-TEM images, Sturm et al. demonstrated that the chain elongation of magnetite particles aligns along the strongest (111) magnetization easy axis[112]. The use of cobalt-doped particles is expected to orient along the [001] axis. In the saturation region, saturation magnetizations are achieved, where all magnetic moments and chains are aligned with the field direction. Upon subsequent reduction of the magnetic field strength to approximately 0.5 T, the magnetic moments initially stabilize due to exchange interactions among themselves, resulting in a minimal

observable reduction in magnetization. Following this, a drop in magnetization occurs, which is now attributed to the separation of the nanochain structures, resulting in the presence of individual nanoparticles at $H < 0.5$ T (Fig. 6a (inlet): transition from phase II to phase I with decreasing $H$). This can also be confirmed by Hu et al., who demonstrated in their work that the high magnetic moment of the nanoclusters makes them highly responsive to a magnetic field, allowing for their alignment in the range of 0.4–5 mT[39].

In the saturation region of the aqueous samples, $M_S$ values of 55 Am² kg⁻¹ to 90 Am² kg⁻¹ are achieved, depending on the heating temperature, the filling volume, and the thickness of the silica precursor (see Supplementary Fig. 8.1). With increasing reaction temperature, the saturation magnetization increases slightly (see Supplementary Fig. 8.1a, 8.1b). This can be attributed to the crystallinity of the nanoparticles, which increases with temperature, resulting in higher $M_S$ as long as no magneto-crystalline stress due to surface defects occurs[57]. However, the diameter of these samples of series 1 and series 2 increases in the order CF-S1-1 < CF-S2-1 < CF-S1-2 < CF-S2-2, showing no significant trend with $M_S$ (see Supplementary Fig. 8.1a).

The samples of series 3 prepared with the precursor with a silica layer exceeding 4 nm exhibit an increasing $M_S$ value with increasing diameter.

An exception is CF-S3-6, as this sample has a significantly higher cobalt content, which is associated with lower $M_S$ values due to the lower $M_S$ of bulk cobalt ferrite (85 Am² kg⁻¹) compared to bulk magnetite (100 Am² kg⁻¹). To reduce the surface energy in the medium, nanoparticles exhibit a higher degree of surface spin disorder compared to the bulk material, resulting in $M_S$ values of approximately 85 Am² kg⁻¹ for cobalt ferrite[1,111]. With the approximation of $M_S = M_S^{bulk} \cdot [(r_P - d_L) \cdot r_P^{-1}]^3$, where $r_P$ describes the particle radius and $d_L$ the thickness of the misaligned layer, $d_L$ is typically on the order of 1 to 2 nm[32,113]. Nevertheless, Gerina et al. calculated the volume of the magnetized and non-magnetic parts of cobalt ferrite samples with different physical sizes, unveiling the size dependence of surface misalignments[114]. As the diameter of the nanoparticles increases, the surface-spin misalignment contribution increases, resulting in higher degrees of non-magnetic volume at the highest examined magnetic field strength of 1.34 T[114]. They also showed that an increase in $H$ reduces the thickness of the unmagnetized layer for all particle sizes, but increases the disorder energy required to polarize the surface spins. However, the surface contribution is strongly temperature-dependent, reaching zero at room temperature measurements[113,115]. Thus, the nanoparticle suspensions with larger diameters approach the saturation value of bulk cobalt ferrite $M_{S,CF}^{bulk}$, which can be observed for the nanoparticles prepared from the precursors with a greater silica thickness (see Supplementary Fig. 8.1c). The higher iron content of the cobalt-doped ferrites increases the saturation magnetization values, which can reach 100 Am² kg⁻¹ for bulk magnetite[116]. This is also the case for magnetite-like particles with a low cobalt content in the CF-A samples (see Supplementary Fig. 8.1c).

If the aqueous samples are compared at different temperatures and filling volumes, the remanence $M_R$ ranges from 1 Am² kg⁻¹ to 34 Am² kg⁻¹ and the coercivity $H_C$ from 1 mT to 4.5 mT for all samples (Fig. 6b). The squareness ratios $M_R/M_S$ and coercivities depending on the cobalt content can be reasonably well described using a parabolic fit, regardless of the temperature or filling volume. Both properties seem to tend to be higher when the temperature is kept low or the filling volume is high within a cobalt ratio of 0.183 ± 0.003. Moreover, magnetite-like particles with low Co/Fe ratio values ($\kappa < 0.18$) exhibit low $M_R$ and $H_C$ at room temperature. By increasing the Co/Fe ratio up to 0.18, the 'hardness' of the particles increases, and the squareness ratio $M_R/M_S$ reaches a maximum at this point. Due to the increasing diameter and further increasing Co/Fe ratio with thicker silica layers of the precursors, as well as increasing saturation magnetization, the $M_R/M_S$ decreases from this point. It should be noted that the maximum $M_R/M_S$ value for randomly oriented, uniaxial systems is 0.5[117]. Considering single-domain particles with cubic magneto-crystalline anisotropy ($K > 1$) that do not interact with each other, this value can reach 0.83[117]. In all cases, the $M_R/M_S$ values obtained in this work remain below this maximum value, which is influenced by factors such as particle size and standard deviation. As the $M_R/M_S$ values range between 0.1 and 0.4, the material's domain state can be considered as pseudo-single-domain for all samples. Furthermore, it is essential to note that the remanence is highly sensitive to interaction effects[117], and the current structures do not constitute a non-interacting system. Based on the determined coercivity values, the particles can be classified as soft magnetic material with $H_C < 51$ mT[118–121]. Initially, $H_C$ increases with rising Co/Fe ratio and diameter, a phenomenon well-documented in the literature[122–124]. As the diameter (or Co/Fe ratio) of the samples increases to values above 40 nm, $H_C$ decreases again, which can be explained by the presence of domain walls expected during the transition from the single-domain to the multi-domain range at a diameter of 40 nm in CF with a maximum $H_C$ of 530 mT[125]. Interestingly, all $H_C$ values do not exceed 4.5 mT. This value is much lower compared to values reported for CF particles within the specified size range and stoichiometric ratio within the literature, where $H_C$ values ranging from 55 mT to 120 mT are reported[120,124,126]. As the aqueous samples showed partial sedimentation after the measurement, the nanoparticles were stabilized in glycerol. Furthermore, these samples and the data obtained offer comparability with the small-angle X-ray scattering measurements, which were only carried out with an 80 wt.% aqueous glycerol mixture (see Supplementary Fig. 8.2).

By dispersing the nanoparticles in an 80 wt.% aqueous glycerol mixture, the hysteresis curve still exhibits a splitting, exemplarily shown for CF-S2-1 in Fig. 6a (bottom). However, the measurements no longer show significant jumps at 1.5 T under increasing field strength and 0.5 T under decreasing field strength. With increasing viscosity, the rotational motion of individual particles, and especially that of nanochains, is hindered. The viscosity of the medium hinders the aggregation of single-domain nanoparticles into nanoparticle chains, requiring higher magnetic field strengths to orient them in the direction of the magnetic field. Individual particles are also affected, although their spin movement is only slightly impaired. In contrast to the aqueous samples, linear correlations of $M_R/M_S$ and $H_C$ in dependence on $\kappa$ (Fig. 6d) as well as the diameter (Fig. 6c) can be recognized. The sample synthesized with pure akaganeite ($d_{TEM} = 25.7$ nm, $\kappa = 0.163$) has a $M_S$ of ~ 50 Am² kg⁻¹, increasing to ~ 72 Am² kg⁻¹ for the sample with PAA ($d_{TEM} = 18.7$ nm, $\kappa = 0.180$) (Fig. 6f). Afterward, the $M_S$ decreases to values of approximately 20 Am² kg⁻¹, as the use of a silica shell influences the crystal phase in such a way that particles with cobalt-rich cores and iron-rich shells are formed. The thickness of this magnetite-like shell is thinner for smaller particles, so that a maximum of $M_{S,CF}^{bulk}$ is assumed. Furthermore, the presence of akaganeite during the particle growth can lead to maghemite formation on the surface, which would result in a reduction of the $M_S$ values. Due to the previously described increase in diameter with the thickness of the silica shell, the $M_S$ values increase again to values up to 87 Am² kg⁻¹ and, in consequence, $M_R/M_S$ does. Overall, the $M_R/M_S$ values range between 0.4 and 0.5, indicating that the particles are close to the single-domain range. These values are slightly higher than in water, which is not surprising, as the interparticle interaction will decrease due to the higher viscosity, thereby approaching the maximum theoretical value of 0.83. $H_C$ values ranging from 3 mT to 5 mT exhibit the same trend, and the particles can still be classified as soft magnetic material. When examining the dependence on $\kappa$, the trends of $M_S$, $M_R$, and $H_C$ are comparable.

By plotting the differential magnetizations against $H$, graphs are obtained as shown in Fig. 6e. The stronger the magnetic moments of a particle can respond to the magnetic field, the steeper the slope of the hysteresis curve in the linear region of the expression $M = \chi_P \cdot H$[127]. This gives rise to the static magnetic susceptibility $\chi_P$, which can be recognized from the plot of $dM/dH$ at $H = 0$ (Fig. 6f and g). It is clearly evident that the static $\chi_P$ is also dependent on the size of the particles and the Co/Fe content, exhibiting the same trend as $M_S$. Furthermore, the magnetic domain size $d_{VSM}$ can be expressed through $dM/dH_{H=0}$ using the following Eq. 1[128]

$$d_{VSM} = \sqrt[3]{\frac{18 \cdot k_B T}{\pi \cdot \rho \cdot M_S^2} \cdot \left(\frac{dM}{dH}\right)_{H=0}} \qquad (1)$$

with $k_B$ as the Boltzmann constant ($1.38 \cdot 10^{-23}$ J K⁻¹), $T$ as the temperature (293.15 K), and $\rho$ as the density of the cobalt ferrite sample (5.30 kg m⁻³)[128]. As summarized in Fig. 7a, the domain size of the particles decreases slightly with the PAA addition and increases again with the silica thickness. The difference between $d_{TEM}$ and $d_{VSM}$ is most likely due to the presence of a magnetically dead layer on the surface of the nanoparticles. As previously stated, an increase in the diameter of the nanoparticles leads to a heightened contribution of surface-spin misalignment, which in turn results in greater amounts of non-magnetic volume. This could provide an explanation for the decrease in the domain size of the particles to about 13 nm when the silica layer exceeds 4 nm, even though the particles have a morphological size of approximately 35 to 40 nm. The crystallite sizes obtained from WH and DS plots are also comparably large (Fig. 7b), indicating that altered crystallinity is unlikely to have an influence. Within this size range, there seems to be a shift in the classification of the particles from the single-domain range to the pseudo-single-domain (or possibly even multi-domain) range. This observation aligns with other studies that show this transition around 40 nm.

As previously mentioned, magnetic self-assembly processes can be described in terms of magnetic interactions[25,107,108]. In the absence of a

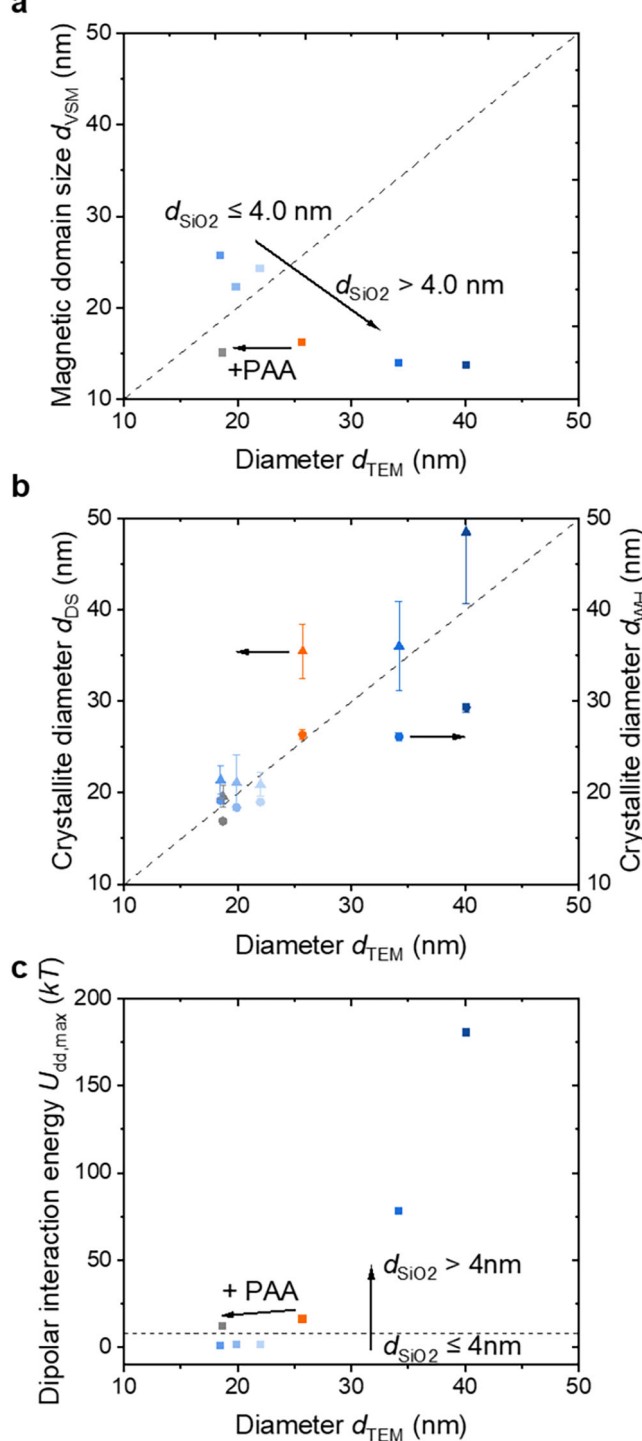

**Fig. 7 | Effect of the silica thickness on the magnetic and crystallite sizes, as well as the magnetic interaction of the samples used for SAXS measurements obtained through TEM, XRD, and VSM. a** The domain sizes (squares) of the particles synthesized without a silica layer are smaller than the sizes obtained *via* TEM (aka: orange, aka@PAA: grey). With a silica thickness of up to 4 nm (color changes from bright blue to dark blue with increasing silica shell thickness of the precursors), the magnetic domain sizes $d_{VSM}$ seemed to be larger compared to $d_{TEM}$, indicating that they are overestimated. When the silica layer exceeds 4 nm, the particles are categorized into the pseudo-single-domain or multi-domain range, resulting in a decrease in the magnetic domain size to 13 nm due to the presence of domain walls. **b** The crystallite size and morphological size of the particles are comparable. There are only slight deviations for the larger particles that indicate differing trends depending on the method of determination (Williamson-Hall: circles, or Debye-Scherrer: triangles). **c** The size-dependent magnetic interaction is depicted by calculating the dipole-dipole interaction energy (squares) at contact, which increases significantly for particles where the silica coating exceeds 4 nm during synthesis.

and is illustrated in Fig. 7c. For the reference particle suspensions, a value of approximately $12 - 16\,kT$ is obtained, which would allow for a "head to tail" dipole configuration (exceeding $8\,kT$) in dilute solutions. The samples synthesized with a silica coating $\leq 4$ nm yield values ranging from $0.7-1.5$ $kT$, placing them in a range to overcome the thermal energy of $1\,kT$ where no chain-assembly of the particles is expected due to insufficient dipole-dipole interactions. This is also evidenced by the lack of splitting in the VSM curves and the TEM images. The larger particles ($d_{SiO2} > 4$ nm) exhibit a significantly increased $M_S$, thereby demonstrating chain assembly across multiple nanoparticles as visualized by TEM. It is important to note that at such high $U_{dd,max}$, particle agglomeration would also be observable, provided that stabilizing effects through repulsive dipole-dipole interactions or electrostatic stabilization *via* ligands are not ensured[129]. Furthermore, the aggregation of nanochains can occur by attractive interaction between antiferromagnetic coupling of antiparallel ordered chains or a gradient of interaction, which may result in the formation of bundles and network-like structures[25]. In summary, the magnetic behavior of the nanoparticles can be described by an interplay of the cobalt content, size, surface disorders, and arrangement of the nanoparticles.

**Cluster formation of magnetic nanochains in dependency on the magnetic field strength**

The alignment of the magnetic moments of the particles in the presence of a magnetic field can be described by stronger dipole-dipole interactions, resulting in an interaction energy scale of $r_i^{-3}$ at all interparticle distances. Because of the magnetic coupling of the magnetic moments, a head-to-tail configuration of the nanochains should be present, separated by a chain distance due to repulsive interactions of the magnetic moments along neighboring chains. *E.g.*, Toulemon et al. were able to assemble iron oxide nanoparticles on a gold substrate using a click reaction while applying a magnetic field, resulting in an interchain distance of 70–80 nm, where the chains consist of several small subunits made up of 6 to 8 particles[25]. The visualization of the chain formation and cluster size of the structures as a function of an applied magnetic field strength in this work was achieved through the analysis of small-angle X-ray scattering (SAXS) data, depending on the field direction, perpendicular, parallel, and overall angles averaged (Fig. 8 (inlet), see Supplementary Note 9.1, Supplementary Figs. 9.1, 9.2, Supplementary Table 9.1). The SAXS data were fitted using the Beaucage model[130–132], which Paula et al. had already used for magnetic nanoparticle clusters[133]. In this case, the intensity can be described by Eq. 2:

$$I(q) = G_1 \exp\left(\frac{-q^2 R_{g_1}^2}{3}\right) + B_1 \exp\left(\frac{-q^2 R_{g_2}^2}{3}\right) h_1^{-P_1} + G_2 \exp\left(\frac{-q^2 R_{g_2}^2}{3}\right) + B_2 h_2^{-P_2}$$

(2)

with $h_1 = \dfrac{q}{\left(erf\left(q\left[\frac{R_{g_1}}{\sqrt{6}}\right]\right)\right)^3}$ and $h_2 = \dfrac{q}{\left(erf\left(q\left[\frac{R_{g_2}}{\sqrt{6}}\right]\right)\right)^3}$.

magnetic field, the magnetic moments of a magnetic particle can align with the magnetic moments of neighboring particles at small separations, resulting in an "in-line" configuration of the dipoles[108]. The dependence of the strength of the dipolar interaction can be described by the Keesom potential, which determines the energy of the interaction[108,111]. This energy scales with $r_i^{-3}$ caused by chain alignments, where $r_i$ denotes the interparticle distance and transitions to $r_i^{-6}$ at larger separations caused by non-magnetic Van der Waals interaction without applying a magnetic field. The maximum dipole-dipole energy at contact, $U_{dd,max}$, describes the magnitude of the magnetic interaction represented as $1/9 \cdot \pi \cdot \mu_0 \cdot (d/2)^3 \cdot M_S^2$, which has been calculated for each particle suspension with their respective $M_S$ value

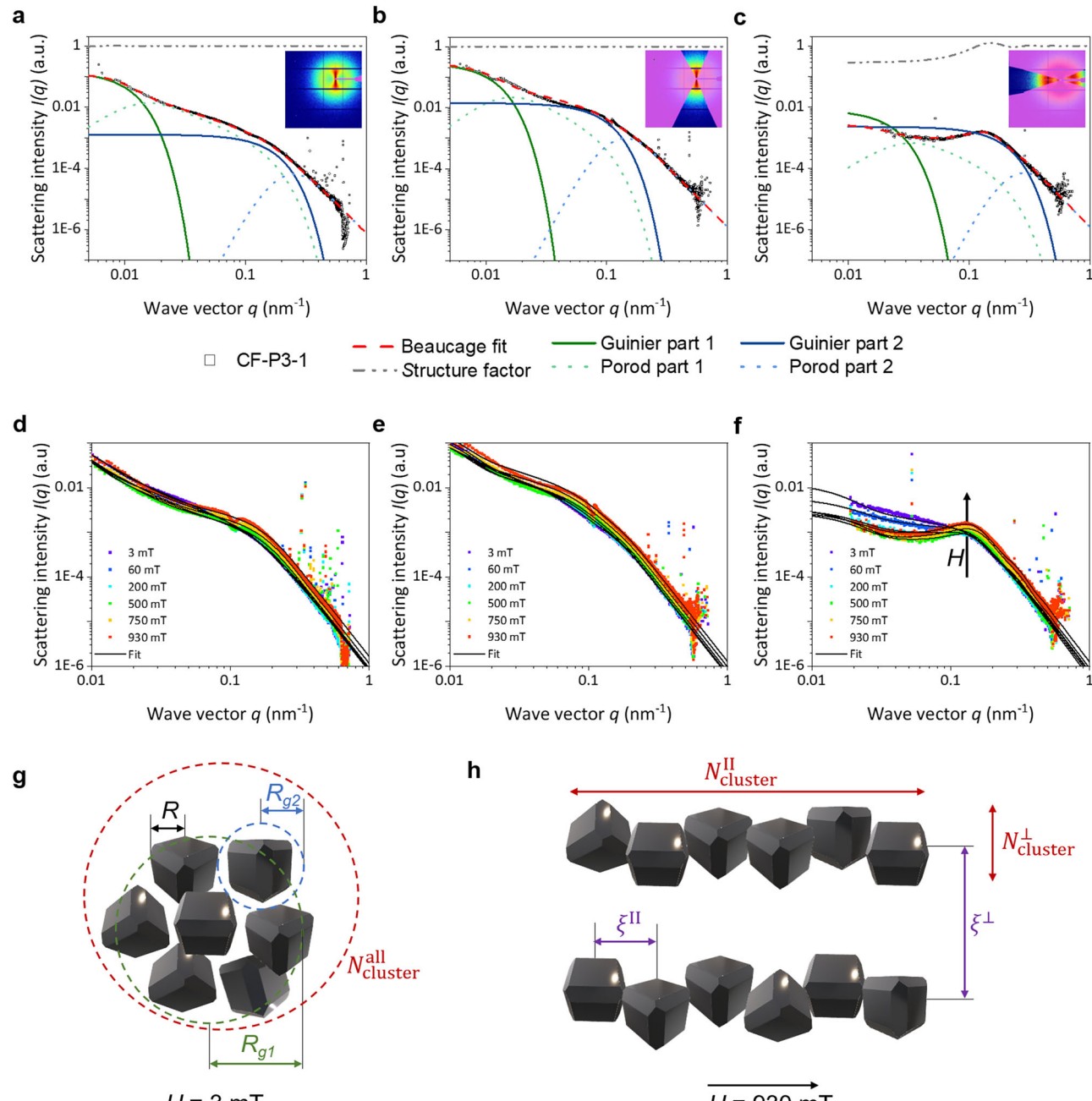

**Fig. 8 | Analysis of the nanoparticle arrangements in dependence on the magnetic field strength.** The scattering intensities (hollow squares) and Beaucage fit (red dashed line) results are depicted for sample CF-P3-1, where the cluster ($G_1$: green line, $P_1$: green dashed line) and single nanoparticle ($G_2$: blue line, $P_2$: blue dashed line) contributions are described by the Guinier and Porod parts, respectively. The fits are calculated **a** for the lowest field strength of approximately 3 mT, averaged over all angles, and for the highest field strength of approximately 930 mT **b** perpendicular, and **c)** parallel to the magnetic field. The 2D SAXS patterns (inlets) are presented with the respective applied masks. The scattering curves and Beaucage fits are shown for different magnetic field strengths **d** averaged over all angles, **e** perpendicular to the field, and **f** parallel to the field. The increasing influence of the structural factor (grey dashed line) is demonstrated by rising magnetic field strength. The schematic representation of the nanoparticle spatial structures elucidates the dependence of the cluster size on the magnetic field strengths of **g** 3 mT and **h** 930 mT visualized for the fits given in Fig. 8a and Fig. 8b, c, respectively. The mean distance of the clusters ($\xi_\perp$) perpendicular to the field remains nearly constant as the field strength increases. In contrast, the length of the clusters increases significantly with the number of particles arranged parallel to the field ($N_{cluster}^\parallel$).

Here, *erf* is the error function. Guinier's pre-factors are represented by $B_1$ and $B_2$, and Porod's by $G_1$ and $G_2$. $P_1$ and $P_2$ are the indices of the power law. The exponential pre-factor $\exp\left(\frac{-q^2 R_{g_2}^2}{3}\right)$ describes the small-scale structural limit[130]. Index 1 belongs to the primary clusters of some NPs, and index 2 is related to the isolated NPs. The wave vector $q$ (Eq. 3) is defined as

$$q = \frac{4\pi}{\lambda} \sin\theta \tag{3}$$

with $2\theta$ as the angle between the incident beam and the scattered beam with the wavelength $\lambda$.

To describe the particle correlation, an additional multiplicative term $S(q)$ has to be considered with[133]

$$S(q) = \left(1 + k\left(3\frac{\sin(q\xi) - (q\xi) \cdot \cos(q\xi)}{(q\xi)^3}\right)\right)^{-1} \quad (4)$$

$S(q)$ is the so-called structure factor. The degree of the correlation is represented by $k$ and $\xi$ describes the interparticle distance.

The calculated contributions of the Guinier (cluster) and Porod (single particle) parts of the Beaucage fits according to Eq. 2 are summarized in Supplementary Fig. 9.3–9.9 for five samples with different silica thicknesses, as well as the comparison samples without and with PAA of series 3. An exemplary analysis with the different contributions of the Beaucage fits is presented in Fig. 8a–c for sample CF-P3-1 synthesized with the akaganeite precursor stabilized with PAA. The average scattering intensities are averaged over all field directions at 3 mT (Fig. 8a), averaged over the section parallel to the field at 930 mT (Fig. 8b), and averaged over the section perpendicular to the field at 930 mT (Fig. 8c). The inlet shows 2D SAXS pattern with the applied masked for each analysis.

Since the field strength was adjusted by varying the distances between the magnets (see Supplementary Table 9.1), SAXS measurements dependent on the magnetic field strength were performed in the range between 3 mT and 930 mT, and the corresponding Beaucage fits at different field directions were calculated as well (Fig. 8d–f). It is clearly evident that the particle correlation increases significantly with rising field strength for this sample. This is associated with an increasing structural factor $S(q)$, which is recognizable as maxima in the SAXS data occurring parallel to the field direction (Fig. 8f).

Furthermore, the Beaucage fit provides access to the radius of gyration $R_g$, the log-normal polydispersity index $\sigma$, and the exponent associated with the interfacial regime[132]. Following that, the median particle radius $R_m$ can be calculated by Eq. 5 through

$$R_m = \sqrt{\frac{5}{3}}R_{g_2}\exp(-7\sigma^2) \quad (5)$$

with the polydispersity index of individual spherical particle sizes by

$$\sigma = \sqrt{\frac{1}{12} \cdot \ln\left(\frac{B_2 R_{g_2}^4}{1.62 G_2}\right)}. \quad (6)$$

Assuming a log-normal size distribution, the mean particle diameter $d_{SAXS}$ is then described with

$$d_{SAXS} = 2 \cdot R_m \exp\left(\frac{\sigma^2}{2}\right). \quad (7)$$

Using the obtained values of the cluster and single particle contributions, the particle sizes, the particle correlations, and the field directions, a schematic illustration of these fits can be visualized for low $H$ averaged over all angles and at high $H$ perpendicular and parallel to the magnetic field in Fig. 8g and h, respectively. A number of cluster particles can be calculated from the fitted values via:

$$N_{Cluster} = \left(\frac{R_{g1}}{R_{g2}}\right)^{P_1} \quad (8)$$

This results in a number of clustered particles $N_{cluster}^{all}$ in all angular directions, as illustrated in Fig. 8g. When an external field is applied, the randomly distributed particles tend to orient themselves into nanochains, enabling the determination of the number of particles in both parallel and perpendicular spatial resolutions, $N_{cluster}^{\parallel}$ and $N_{cluster}^{\perp}$, respectively (Fig. 8h).

These values correspond to the length and width of such nanochains or bundles, where multiple chains aggregate together. The possibility of comparing the aggregate size with $N_{cluster}^{all}$ arises from the product of the parallel and perpendicular measurements, with an approximation to the visualization above all azimuthal angles. The calculated cluster numbers via Eq. 8 (Fig. 9) are depicted for the magnetic field strengths that were considered interesting in the VSM measurements. These included the smallest (3 mT) and highest (930 mT) field strengths, as well as the strength at which a drop due to nanochain reorientation was observable (500 mT). Since the magnetic domains were determined using VSM measurements of the glycerol solutions, which were utilized at the same concentration for SAXS measurements, a ratio of morphological and magnetic diameters was established. The cluster numbers were then specified as a function of the size ratio $d_{TEM}/d_{VSM}$. A more detailed representation of the cluster numbers in dependence on the field strength, additionally with decreasing field strengths, is provided in the Supplementary Fig. 9.3i–9.9i and Supplementary Fig. 9.10.

First, it should be noted that the cluster sizes, which are averaged over all azimuthal angles, result in smaller particle counts than those derived from the product of the length and width of the chains (see Supplementary Fig. 9.10). Figure 9a demonstrates that $N_{cluster}^{all}$ is maximized at the lowest $H$ for the control sample without PAA (CF-A3), reaching a count of 1000 randomly distributed particles in a cluster. $N_{cluster}^{all}$ for all other samples is observed to be lower than that of CF-A3. Since CF-P3-1 contains the smallest particle diameters, the energy of dipolar interactions is lower compared to larger particles, but the higher $M_S$ results in a $U_{dd,max}$ of 12 $kT$, consequently facilitating this aggregation behavior. Additionally, it is important to consider that the presence of citrate on the surface imparts a negative surface charge, leading to repulsive forces that effectively separate the particles, thereby inhibiting unwanted agglomeration into random clusters of larger scale. Because of the higher surface-to-volume ratio of smaller particles compared to the particles synthesized with silica, this electrostatic repulsive effect should be increased. This effect, however, is opposed by the attractive interactions that can arise from the non-magnetic Van der Waals (scales linear with the diameter) and magnetic dipole-dipole forces (scales linear with the volume). The magnetic particles act as small magnets due to the attractive interaction of their magnetic dipoles, which increases with the particle diameter (Fig. 7c). If one assumes that there is no alignment of the particles into chains at the lowest applied $H$ of 3 mT, one would anticipate an increase in cluster size due to the enhanced magnetic interactions when increasing $d_{SiO2}$ (and $d_{TEM}$, respectively) as demonstrated in Fig. 9a and in the Supplementary Fig. 9.10.

As $H$ increases, $N_{cluster}^{all}$ exhibits a slightly decreasing trend for sample CF-A3, which suggests an decrease in dipolar interactions that weaken at greater separations but is still present in the scale of $r^{-3}$[108]. This observation is noteworthy, as a similar behavior is observed upon subsequent reduction of $H$, indicating a lack of reorientation into a random system (Fig. 9b, see Supplementary Figs. 9.10). This phenomenon is probably associated with a partial irreversible agglomeration of the particles. Conversely, the sample containing PAA reveals a pronounced increasing trend in cluster numbers over all angles with an increase in $H$, followed by a subsequent decline when $H$ is lowered. $R_{g1}$ also increases with rising magnetic field strength, indicating an increase in the number of particles in the cluster. As these particles have the smallest diameter, the particle cluster number is higher than for larger particles. The particles can approach each other over shorter distances, as the repulsive forces will be weaker. Interestingly, the particles synthesized with akaganeite@PAA@silica precursors can be classified into two ranges, namely above and below $d_{SiO2}$ of 4 nm. The particles in both categories show no significant change in $N_{cluster}^{all}$ with the field strength. When the field strength is reduced again up to 3 mT, the distribution of $N_{cluster}^{all}$ is comparable to the initial observations. Only the sample CF-A3 shows a slight decrease, which is likely related to the irreversible agglomeration described above.

The evaluation of particle assembly into nanochains and bundles was conducted based on $N_{cluster}^{\parallel}$ and $N_{cluster}^{\perp}$ with respect to their length and

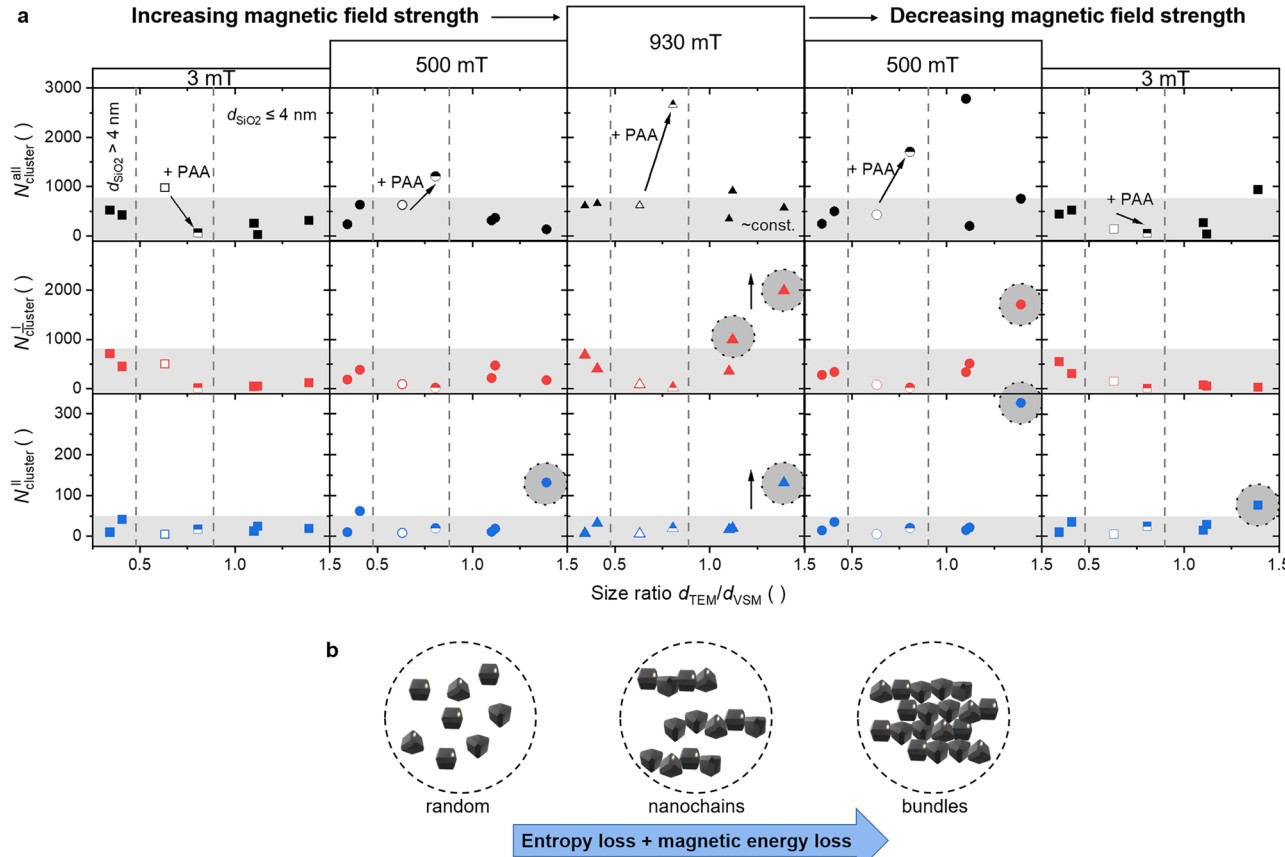

**Fig. 9 | Clustering of the nanoparticles in nanochains and bundles, and re-clustering of the particle agglomerates. a** The particle numbers of the clusters (aka: hollow, aka@PAA: half-filled, aka@PAA@SiO$_2$: filled) are depicted in dependency on the ratio of morphological and magnetic domain size and on the magnetic field with increasing as well as decreasing field strength for the average of all field directions ($N_{cluster}^{all}$, black squares), averaged angles in perpendicular field direction ($N_{cluster}^{\perp}$, red squares), and averaged angles parallel to the field direction ($N_{cluster}^{\parallel}$, blue squares). $N_{cluster}^{all}$ increases to values of approx. 2500 NPs with increasing field strength up to 930 mT for the smallest particles of the reference sample CF-P3-1. With decreasing field strength up to 3 mT, $N_{cluster}^{all}$ decreases to values of approx. 80 NPs for the smallest particles of this sample. The values of $N_{cluster}^{\parallel}$ are nearly constant for each sample, except for CF-S3-5. Here, the cluster number increases (decreases) with increasing (decreasing) magnetic field strength. However, $N_{cluster}^{\perp}$ reach significantly higher values with increasing field strength for the samples synthesized with a silica layer, respectively $d_{TEM}/d_{VSM} > 1$. **b** The size- and concentration-dependent formation of nanochains and bundles is due to an entropy and magnetic energy loss. The NPs orient in nanochains due to increasing dipolar interactions with increasing diameter and field strength, resulting in a magnetic energy loss. The formation of ordered systems, such as nanochains and bundles, is also accompanied by an entropy loss.

width as well as the interparticle distances. For all particle samples, $N_{cluster}^{\perp}$ is consistently greater than $N_{cluster}^{\parallel}$, indicating an arrangement of short chains that is oriented parallel to the magnetic field, which is applied perpendicular to the X-ray beam, which was also demonstrated by Toulemon et al.[25] It is first evident that CF-A3 exhibits a decreasing trend in $N_{cluster}^{\perp}$ from 500 to 80 particles as the magnetic strength increases. This trend is consistent with the observations regarding $N_{cluster}^{all}$. Since $N_{cluster}^{\parallel}$ shows constant values of approximately 5, it appears that particles from longer aggregates are assembling into shorter chain-like structures. However, one should not overlook the correlation degree, which does not exceed 1 in this case (see Supplementary Fig. 9.3). CF-P3-1 shows a slightly higher degree of correlation, which increases to over 2, especially at high magnetic field strengths (see Supplementary Fig. 9.4). Here, $R_{g2}$ of the single particle in the perpendicular direction indicates a size of approximately 22 nm, while in the parallel direction it is around 10 nm. This could suggest an anisotropic orientation of the particles themselves due to the slight anisotropy of the particles. Since both cluster numbers, $N_{cluster}^{\parallel}$ and $N_{cluster}^{\perp}$, are similarly small, ranging from 16 to 25, a spherical aggregation of smaller sub-units composed of two particles may have occurred. The behavior of both samples shows no clear magnetic field dependence when evaluated in both perpendicular and parallel field directions.

In contrast, the samples synthesized with silica precursors exhibit a different behavior in perpendicular and parallel field directions. As previously mentioned, these samples can be categorized into two categories, one with a ratio of $d_{TEM}/d_{VSM}$ below 0.5 and the other above 1.0. The first range arises from the significantly smaller magnetic domain size of approximately 13 nm for much larger particles over 30 nm, resulting in a likely much higher proportion of the magnetically dead layer. Here, the correlation degree is small and independent on the magnetic field strength (see Supplementary Fig. 9.8g and Supplementary Fig. 9.9g). $R_{g1,\perp}$ is calculated to be an average of 300 nm, while $R_{g1,\parallel}$ is 100 nm (see Supplementary Fig. 9.8d and Supplementary Fig. 9.9d). This behavior suggests a tendency toward bundle formation. There are no significant differences in field-dependent behavior between $N_{cluster}^{\parallel}$ and $N_{cluster}^{\perp}$. Compared to the other ratio category, it is observable that the cluster numbers initially decrease until 0.5 T and then rise again (see Supplementary Fig. 9.8k and Supplementary Fig. 9.9k). This trend is also evident with decreasing field strength. This behavior can be linked to the splitting of the VSM curves at H < 0.5 T, where a drop occurs. This is attributed to the alignment and the additional high magnetic moments of the chains. Furthermore, the particles with larger sizes exhibit a dramatically higher dipole-dipole interaction energy, which could lead to the formation of nanochains and/or bundles even at higher separations of the single particles.

The second category, consisting of the samples CF-S3-1, CF-S3-3, and CF-S3-5, is clearly distinguished by an increased correlation degree with magnetic field, reaching up to 8 (see Supplementary Fig. 9.5–9.7). The cluster numbers $N_{cluster}^{\perp}$ show an increasing trend with rising $d_{TEM}/d_{VSM}$. Furthermore, the $N_{cluster}^{\perp}$ values of these three samples also increase with increasing field strength. For the first two samples, the chain width at 930 mT ranges from 360 to 1000 particles, while $N_{cluster}^{\parallel}$ remains relatively constant at around 8 to 24. Since $R_{g1}$ indicates a size of 300 nm x 100 nm or 200 nm x 75 nm, this suggests the formation of bundles, also verified by a decreasing interparticle distance in the perpendicular direction $\xi_{\perp}$ with the field strength (see Supplementary Fig. 9.5h and Supplementary Fig. 9.6h). For sample CF-S3-1, *e.g.*, $\xi_{\perp}$ decreases from 800 nm to around 55 nm when the magnetic field strength exceeds 100 mT. This effect is justified by the significantly larger magnetic domain size, which accounts for the entire particle volume within the single-domain range. Consequently, the subunits aggregate into larger complexes. The most interesting sample (grey circle) appears to be CF-S3-5, which, alongside the field dependence of the $N_{cluster}^{\perp}$ values, also shows a strong increase in $N_{cluster}^{\perp}$. This increase can be observed from 100 mT (110) to 500 mT (132), reaching a value of 126 at 930 mT (see Supplementary Fig. 9.7). With $N_{cluster}^{\perp}$ of 305 at 100 mT in parallel, it appears that chain-like assembly is already occurring. The interparticle distance $\xi_{\perp}$ now decreases for higher $H$ values, indicating that particles are assembling more closely together (see Supplementary Fig. 9.6). This is further reflected in the decrease of $N_{cluster}^{\perp}$ to as low as 174. Upon a further increase in magnetic field strength, $N_{cluster}^{\perp}$ rises again to 2000, significantly exceeding the values of $N_{cluster}^{all}$, suggesting the occurrence of bundle clustering. Interestingly, this seemed to be reversible by decreasing the field strength.

The particle numbers in dependency on $M_S$, $M_R/M_S$, and $H_C$ are summarized in the Supplementary Figs. 9.11, 9.13, respectively. $N_{cluster}^{all}$ increases to values of approx. 2500 NPs with increasing field strength up to 930 mT for the smallest particles with comparable high $M_S$ of 70 Am$^2$kg$^{-1}$, showing a random orientation as the numbers for perpendicular and parallel orientation are comparably low. The values of $N_{cluster}^{\parallel}$ are nearly constant for each sample, except for CF-S3-5 with the lowest $M_S$. These particles can probably be most easily aligned using an external field. However, $N_{cluster}^{\perp}$ reaches significantly higher values with increasing field strength for the samples synthesized with a silica layer smaller than 4 nm, possessing moderate $M_S$ values. This behavior is confirmed in the $M_R/M_S$ and $H_C$ representation, where the sample with the lowest values shows the highest $N_{cluster}^{\perp}$ and $N_{cluster}^{\parallel}$. Only the smallest particle sample synthesized with PAA exhibits higher $N_{cluster}^{\parallel}$, which indicates a random orientation and does not support chain formation based on the evaluated data.

Additionally, the mean particle diameters $d_{SAXS}$ can be determined using Eq. 7 and are illustrated in the Supplementary Fig. 9.14 for all field directions at the lowest and highest applied magnetic field strength in comparison to $d_{TEM}$. The calculated values are comparable in the trend behavior obtained *via* XRD and VSM, although $d_{SAXS}$ is slightly smaller than $d_{TEM}$. At all azimuthal angles, $d_{SAXS}$ at 3 mT and 930 mT correspond well, except for sample CF-S3-3. Since the calculated values are highly dependent on the fitting of the data and are available twice for each respective field strength, albeit with different absorbers (to minimize beam damage), discrepancies may arise.

### Critical assessment of the particle systems regarding their biomedical application potential

**Influence of particle characteristics.** For potential medical cancer treatments, the hyperthermal effect is arguably the most interesting one. As the SAR value is highly dependent on numerous material-related parameters, including size, size distribution, hydrodynamic volume, particle concentration, aggregation behavior, viscosity, and saturation magnetization, relaxation times, among others, the discussion on the influence of each factor is broad[33,52,134–140].

Sathya et al. synthesized cobalt-doped ferrite cubes with edge lengths of 14–27 nm and $x$ ranging from 0.1 to 0.7 *via* thermal decomposition, with the highest SAR values observed for particles measuring 17–19 nm[124]. These

SAR values were also higher than those of pure iron oxides. Since we are synthesizing particles in the range of 14 to 47 nm, the SAR values are expected to be highest for the smaller particle diameters in our series (see Supplementary Table 4.1). However, it should be noted that the optimal diameter increases with the amplitude used in hyperthermia experiments when the magnetic field amplitude exceeds 2.7 mT (*i.e.*, in the nonlinear region of the magnetic response to the applied magnetic field described by the Stoner-Wohlfarth model)[137]. In conclusion, no general statement can be made regarding the particle diameter obtained in this work. Additionally, the arrangement of nanochains is expected to significantly influence the heating performance of the suspension, as discussed later.

It has also been demonstrated that increased polydispersity in particle size has an impact on hyperthermia, typically resulting in a decrease in efficiency[138]. The particles synthesized here exhibit size standard deviations ranging from 18% to 35%, resulting in values ranging from 0.04 to 0.13 (calculated *via* Supplementary Equation 4.4[141]), except the outliers CF-S1-1 and CF-S3-1 (Supplementary Table 4.2). These values are comparable to those reported for aqueous coprecipitation reactions (10–30% or PDI = 0.01–0.09, respectively), even though they appear relatively high in absolute terms. Furthermore, the use of PAA or silica in the reaction solution does not result in a clear trend in dispersity. However, an average silica thickness of around 4 nm seems to have a positive influence on dispersity, exhibiting the lowest PDI.

According to Arteaga-Cardona et al., particle aggregation significantly reduces the heating efficiency from 96 W g$^{-1}$ to 12 W g$^{-1}$ (with $f$ = 110 kHz, $H_0$ = 20 mT) for 15 nm cobalt ferrite using either an oleic acid-stabilized system in hexane or a tetramethylammonium hydroxide solution[33]. They demonstrated the best heating performance on particles stabilized in hexane, which limits the implications to a theoretical statement. However, disaggregation reduces the hydrodynamic volume, enabling faster Brownian relaxation. In our case, the particle species with diameters ranging from 14 nm to 18 nm exhibit good colloidal stability, as verified primarily by visual observations of the absence of sedimentation over time. Therefore, these particles relax *via* Brownian motion, depending on their size[142]. For particles of similar size but with low cobalt contents ($x$ = 0.38), we previously reported SAR values of 183 W g$^{-1}$[57]. In contrast, larger particles may sediment or aggregate due to their higher interparticle interactions as shown for pure iron particles, if stabilization is insufficient, thereby reducing their heating performance[137].

Nguyen et al. determined that the diminishing effect of the heating performance is less pronounced for higher magnetic anisotropy ($K$) values compared to smaller $K$ values[138]. They also identified a lower requirement for monodisperse suspensions at higher $K$ values of up to 41 kJ·m$^{-3}$. More specifically, 35% of the heating efficiency is preserved with a fourfold increase in $K$. This could be realized with a higher shape and magneto-crystalline anisotropy, as presented for the cobalt-doped particles within this work, which should be a magnitude higher compared to magnetite. In previous work, we calculated magneto-crystalline anisotropy values of 32 to 59 kJ·m$^{-3}$ for comparable cobalt-doped particles, which places these particles within a similar range[66].

To evaluate the SAR performance under conditions that mimic the cellular environment and aim for clinical use, it is crucial to determine and characterize heating effects at higher viscosities than in aqueous media. The Néel relaxation will be less affected by the biological environment, as it is independent of the viscosity, resulting in a preference for this mechanism in hyperthermia treatments. Therefore, the impact of $M_S$ and $H_C$ on the SAR will be discussed based on the magnetization measurements conducted in 80% glycerol/water mixtures. In general, higher $M_S$ results in higher SAR as predicted by Gavilán et al. and others[44,139]. Considering this, one could conclude that larger particles, due to their higher $M_S$ values as shown in Fig. 6e, should also exhibit higher heating efficiency, provided their stability can be maintained. Additionally, the particles synthesized with aka or aka@PAA precursors also possess high $M_S$ values ranging from 45 to 75 Am$^2$ kg$^{-1}$, which could enable their use in hyperthermal applications. These trends are supported by the data on static susceptibility (Fig. 6e), which

https://doi.org/10.1038/s42004-025-01730-9                                                                                                                         **Article**

increases as the magnetic moments of the particles respond more effectively to the magnetic field.

At low field strengths as provided in medical treatments, it was observed that the very weak frequency dependence of SAR is primarily due to $H_C$[143]. This is because the applied fields were insufficient to overcome the energy barrier for inducing a magnetic moment reversal along the magneto-crystalline easy axis. However, once the field strength exceeded $H_C$, a complete hysteresis cycle occurred, which was associated with a pronounced frequency dependence of hysteresis losses and resulted in an increase in SAR. This behavior was determined in cobalt ferrite particles and directly translated to their larger $H_C$[144]. For example, Phong et al. demonstrated that 13.5 nm CoFe$_2$O$_4$ particles with $H_C$ of 4.3 mT are dominated by Néel and Brown relaxation loss and show a high SAR of 142 W g$^{-1}$, although the $M_S$ is moderate with 39 Am$^2$ kg$^{-1}$[145]. Larger particles with high $M_S$ up to 76 Am$^2$ kg$^{-1}$ showed a significant decrease in SAR to 7 W g$^{-1}$, which was related to the higher $H_C$ values of up to 85 mT in comparison to the applied magnetic field. Since the particle systems in this work do not exceed $H_C$ values of 4.5 mT (Figs. 6c and 5d), the applied fields for hyperthermia treatments (typically <30 mT) should be sufficiently strong to induce a reversal of the magnetic moments. Furthermore, the frequency dependence of hysteresis losses should be significantly more pronounced, which in turn would lead to an increase in SAR[143].

Lastly, the squareness ratio $M_R/M_S$ provides valuable insight into nanostructured systems. It is categorized into ranges less than 0.5, equal to 0.5, and greater than 0.5, which respectively indicate randomly distributed interacting nanoparticles, randomly distributed non-interacting particles, and anisotropic structures resulting from anisotropic nanoparticles or anisotropic orientation within chains[146]. Due to the increasing $M_R/M_S$ with diameter, the larger particles are expected to arrange into more anisotropic assemblies preferentially. However, the differences observed are only marginal, ranging from 0.35 to 0.48. However, the hysteresis curves provided in this work offer only limited insights, especially since solutions rather than solid samples were measured. Since hysteresis cycles are also dependent on the applied magnetic field amplitude, measurements are necessary to identify the magnetic field strength at which chain-like assemblies are favored, resulting in an increase in $M_R/M_S$[146]. However, in this work, the chain-like structure formation, dependent on the field strength, is achieved using the SAXS measurements discussed in the previous section.

**Influence of chain assembly**. Additionally, other mechanisms, such as the reduction of dipolar interactions and the formation of particle chains, may also play a crucial role in enhancing the particles' heating performance[136]. However, as the chain formation process in this work was characterized on permanent magnetic fields, the comparison with data from hyperthermia experiments should be viewed with caution.

An investigation of chain formation in alternating current (AC) fields is warranted, although the experimental implementation is limited by the constraints imposed by AC field hyperthermia setups. Relevant studies have investigated the effects of the chaining process on the hyperthermia performance of magnetite particles of different sizes[146,147]. Morales et al. analyzed the changes in susceptibility and remanence associated with chain formation, finding that these effects were only observed for magnetite particles with a diameter of 34 nm, and not for particles with diameters of 12 nm or 53 nm[146]. The magnetic reversal of the small particles is due to the Néel mechanism, also combined with a small dipolar particle interaction. As a result, the thermal fluctuations exceed the dipolar energy, preventing the formation of chains. The large particles are dominated by Brownian motion, but agglomeration occurs, avoiding a chain arrangement.

The unique ability of CF particles to form nanochains in vivo was experimentally observed by, e.g., Balakrishnan et al.[144] They investigated well-defined structures, comprising a median of 4 nanocubes per chain without hyperthermia treatment or 7 nanocubes per chain with hyperthermia treatment, within endosomal regions of the murine xenograft tumor model using nude mice and human A431 epidermoid carcinoma cells. In contrast to Morales et al., it has been proposed that chain formation

occurs only at significantly larger Néel relaxation times compared to Brownian and diffusion relaxation times, as only under these time conditions the magnetic moments are sufficiently stable[146]. Then, the magnetic dipolar interaction between the particles is sufficiently dominant to induce the assembly into chains. Because CF is dominated by Néel relaxation at smaller diameters compared to magnetite, where this dominance occurs only at larger diameters due to lower $K$, 100% of their size-distributed CF particles (17 ± 2 nm, PDI = 0.08) can contribute to chain formation. In contrast, only 42% of the magnetite particles (18 ± 3 nm, PDI = 0.16) lie within the aggregation region. Additionally, comparing the magnetic coupling strength with $\lambda = \mu_0 \cdot (M_S \cdot V_P)^2/(2\pi \cdot d_{cc}^3 \cdot k_B \cdot T)$ of spherical and cubic particles of the same particle volume $V_P$ and $M_S$ indicates that cubic particles are more prone to chain formation. Here, $d_{cc}$ describes the center-to-center interparticle separation distance. Since the larger diameters in our work exhibit more cubic-like morphologies, the chain formation effect can be attributed to the increased $\lambda$, especially for solvent-driven self-assembly processes, as given by TEM preparation. Even if spherical particle shapes are assumed for all particle systems in this work, $\lambda$ increases for larger particles with higher $M_S$, followed by those synthesized with PAA and bare akaganeite (Fig. 7e). This supports the assumptions derived from the TEM images (Fig. 6a). It is further essential to highlight the significant potential of such CF particles to enhance hyperthermia treatments, and in contrast to cubic iron oxide, enable a substantial delay in tumor growth. Because of the chain formation, the mechanical damage after injection is higher than for unchained particles. Hyperthermia applications with three treatments, as demonstrated in the work by Balakrishnan et al., resulted in tumor reduction within 30 days, no recurrence over 200 days, and disease-free survival of up to 7 months. In comparison, the control group survived a maximum of 30 days.

**Influence on toxicity, cellular distribution, and biodegradation**. Ozer et al. observed that neither bare CoFe$_2$O$_4$ nanoparticles nor ascorbic acid-coated CoFe$_2$O$_4$ particles exhibited significant cytotoxic effects on L929 cells, thus underscoring their biocompatibility and low toxicity[148]. The antioxidant properties of the ascorbic acid (AA) coating, renowned for its capacity to mitigate oxidative stress, may play a crucial role in protecting healthy cells and enhancing their viability compared to uncoated ferrites. The contrasting responses of healthy L929 cells and C6 glioblastoma cells to AA-coated cobalt ferrite systems could be explained by the protective influence of AA against oxidative stress, or alternatively, by its potential to trigger cell death pathways in cancerous cells. Such mechanisms might contribute to increased apoptosis or other forms of cell death selectively in tumor cells.

Regarding the toxicity of the cobalt-doped particles, we would like to reference the systematic work of Flores Urquizo et al., who have demonstrated that bare cobalt-doped particles, especially in the range of $x$ from 0.2 to 0.6, exhibit lower toxicity compared to iron oxides ($x = 0$) and cobalt ferrite ($x = 1$)[59]. Additionally, Albarqi et al. verified that the released amount of Co ions from Co-doped iron oxide nanoparticles after a 7-day incubation period in complete culture medium at 37 °C was only 0.03%[149]. A substantial release of Co ions can therefore be excluded.

In contrast, Balakrishnan et al. demonstrated that the intrinsic toxicity of cobalt ions can be utilized for the simultaneous chemotherapy of cancer cells combined with hyperthermia therapy[144]. This occurs through the cellular degradation of the particles under acidic environmental conditions within tumor tissue, allowing for the slow release of cobalt ions from cubic Co$_{0.65}$Fe$_{2.35}$O$_4$ particles, as also considered minimally invasive by Albarqi et al.[149] Interestingly, they attribute this unique behavior to the chain formation under alternating field observed only in cobalt ferrite, but not in magnetite, as discussed earlier. Consequently, this approach could prevent the use of an additional cytotoxic agent, which is often used in combination with other cancer treatments. Regarding the cellular distribution of these nanocubes, they investigated the particle localization within endosomal regions, a process that can be facilitated by the acidic pH in these compartments. The distribution of the cubes was examined, with observations

limited to the tumor region, and no detectable presence was found in the liver or spleen. However, the size and shape changed to partially reduced and deformed ones. As demonstrated, intratumorally injected CF particles can be eliminated by ferritins, as successfully shown for Fe ions in the spleen[150]. The biodistributed Fe and Co ions are also verified in the tumor and spleen. The first tissue shows a decreasing amount of ions, while the latter shows an increasing amount, concerning the post-treatment time. However, the Co amount is only 17 µg after 80 days, which explains the absence of physical or psychological changes, also demonstrated for the control group injected with an equivalent amount of $CoCl_2$. Interestingly, this group shows no significant tumor regression due to the intrinsic toxicity of the salt itself. Therefore, it can be concluded that the CF particles are needed for a combined therapy with minimal toxicity. This could also apply to the particles investigated in this work. The slightly lower cobalt content might have a positive effect by reducing the toxicity effects to a lower value. In any case, hyperthermal measurements should be conducted in relation to viscosity and salinity, along with corresponding treatments, to assess the actual effect of the synthesized CF nanochains.

## Conclusion

Bioinspired magnetic nanochains such as magnetosomes served as a model for this work. However, it must be emphasized that their application remains highly controversial, primarily because of the significant limitations in the scalability of cultivation and purification processes. Nonetheless, this research area is highly prominent, and synthetic strategies are being developed to replicate these systems and facilitate their practical application. This study presents an innovative and environmentally friendlier method compared to conventional protocols due to the lower cobalt content and aqueous approaches for synthesizing magneto-responsive nanostructures, specifically chain-like arrangements composed of truncated cubic cobalt-doped ferrite particles.

Akaganeite nanorods were effectively functionalized in the first step through surface modification with PAA and subsequent silica encapsulation. The synthesis of functionalized precursors with diverse silica shell thicknesses ranging from 2 nm to 7 nm has been successfully demonstrated *via* a modified Stöber process. The resulting akaganeite nanorods, synthesized under controlled conditions, exhibit well-defined dimensions and favorable stability when subjected to surface modification using PAA. The characterization of these nanorods reveals significant insights into their hydrodynamic properties, with the zeta potential measurements indicating a strong influence of pH on particle stability and agglomeration behavior. The adjustment of PAA concentrations has been shown to play a critical role in modulating both the surface properties and the aggregation of the nanorods, ultimately affecting the thickness of the silica shells formed. By utilizing optimized ratios of TEOS and ethanol, uniform silica encapsulation of the nanorods can be achieved, leading to enhanced control over particle size and distribution. Moreover, spectroscopic analysis confirms the successful surface modification while maintaining the integrity of the underlying akaganeite crystalline structure. This comprehensive approach not only elucidates the synthesis parameters but also highlights the critical interplay between surface modification and silica shell formation, thereby exploring the functionalization potential of these silica-coated nanorods.

The subsequent step addresses the conversion of the modified precursors *via* a hydrothermal reaction. The addition of PAA likely contributes to smaller crystallite sizes and higher cobalt-to-iron ratios than the bare akaganeite precursor. By introducing a silica layer, larger CF particle sizes can be observed with increasing temperature. The variation in silica shell thickness reveals a clear correlation between cobalt content and nanoparticle dimensions. Significant variations in crystallite size were attributed to the thickness of the silica layer of the precursors, highlighting a change in characteristics at a silica thickness of 4 nm. Notably, the transformation mechanisms involved in the formation of magnetite from akaganeite are intricately delayed by the presence of silica. The observed changes in the $2\theta$ angles of the XRD patterns indicate the complexities of crystallization and ion exchange processes occurring during synthesis. As the silica layer

became more substantial, it effectively delayed the transformation of akaganeite, thereby influencing the distribution of cobalt ions throughout the nanoparticles. Moreover, magnetic property assessments revealed significant hysteresis behavior and improved magnetization, reaching 90 Am²/kg for larger nanoparticles, indicating their potential in various applications. Considering the synthesized particle morphology in our work, magnetosome-like structures are primarily observed in larger individual particle diameters. The research underscores the importance of controlling both the silica thickness and the additive concentration to optimize the magnetic properties of the resulting chain-like nanoparticle structures.

## Methods
### Materials

$FeCl_3 \cdot 6H_2O$ (97%) and $FeCl_3 \cdot 6H_2O$ (99%) was obtained from Sigma-Aldrich (St. Louis, MO, USA), $FeCl_2 \cdot 4H_2O$ (97%) from Sigma-Aldrich (St. Louis, MO, USA), $CoCl_2 \cdot 6H_2O$ (97%) from Sigma-Aldrich (St. Louis, MO, USA), $Na_2HPO_4 \cdot 2H_2O$ (99.5%) from Merck (Darmstadt, Germany), sodium salt of polyacrylic acid (PAA, $M_w$ = 2100 g mol⁻¹) from Fluka Inc. (Buchs, Switzerland), 25% ammonia solution from VWR International (Fontenay-sous-Bois, France), (tetraethoxy)silane from Sigma-Aldrich Chemistry (Steinheim, Germany), absolute ethanol (99.8%) from Fisher Scientific (Loughborough, United Kingdom), NaOH (99%) from Grüssing (Filsum, Germany), 25 wt.% tetramethylammonium hydroxide (TMAH) solution from Sigma-Aldrich (St. Louis, MO, USA), citric acid (99%) from Sigma-Aldrich Chemistry (Steinheim, Germany), and trisodium citrate was obtained from Honeywell Burdick & Jackson (Sleeze, Germany). All chemicals were used as received without further purification. Ultrapure water (Milli-Q® quality, resistivity > 18.2 MΩ·cm) was obtained from a Millipore Milli-Q® water purification system from Merck (Darmstadt, Germany). We utilized a system from Berghof Instruments (Eningen unter Achalm, Germany) for the reactions conducted in a hydrothermal reactor. The hydrothermal reactor comprises a Teflon liner with a capacity of 25 mL, along with a PTFE tube for degassing, housed within a stainless-steel autoclave. While the reaction occurred at 160 °C or 190 °C and 10 bar, the pressure reached a maximum of 15 bar.

### Synthesis of the akaganeite precursors

In a 500 mL three-neck flask, a 7.8 mg mL⁻¹ aqueous solution of $Na_2HPO_4 \cdot 2H_2O$ was added to Milli-Q water (18.2 MΩ·cm) to obtain a final concentration of 0.4 mmol L⁻¹ and stirred at 500 rpm for 5 minutes. Then, $FeCl_3 \cdot 6H_2O$ (99%, 0.0203 mol L⁻¹) was added to the stirred solution. The orange mixture was refluxed at 80 °C for two days, evolving into a clear red-brown solution. The mass fraction was 1.1 wt.%. The reaction conditions are presented in the Supplementary Table 10.1.

### Synthesis of the akaganeite precursors modified with PAA

The following syntheses were carried out following a modification of the procedure by Li et al.[69] All reaction parameters for the different batches are summarized in Supplementary Table 10.2.

For aka@0.04PAA, 6.6 mg of PAA (3.1 µmol) was dissolved in 174 mL of water while stirring. The same procedure was employed for aka@0.12-PAA, where 22.8 mg of PAA (10.9 µmol) was used, respectively 243.9 mg of PAA (116.1 µmol) for aka@0.40PAA. Then, 10 mL of the akaganeite solution (1.1 wt.%, 1.24 mmol, 110 mg particle mass) was added. The mixture was stirred for 3 days at 400 rpm and room temperature. Subsequently, the suspension was centrifuged four times at 11,000 rpm (12,851 g) for 15 minutes at 20 °C. The sediment was washed three times with 10 mL of water each time. Finally, the precipitate was dissolved in 1.75 mL of water to obtain a 3.5 mol L⁻¹ particle suspension.

For a fivefold larger scale of the aka@PAA functionalization, 367.8 mg of PAA (0.1751 mmol) was dissolved in 870 mL of Milli-Q water using a large oval stir bar. Subsequently, 50 mL of the akaganeite solution (1.1 wt.%, 6.19 mmol, 550 mg particle mass) was added under stirring. The reaction mixture was stirred at 500 rpm for 69 hours. The reaction mixture was concentrated to 70 mL and then centrifuged once at 11,000 rpm (12,851 g)

for 45 minutes at 20 °C. The precipitate was washed thrice with 20 mL of water each at 11,000 rpm (12,851 $g$) and 20 °C for 15 minutes. Subsequently, it was dissolved in 8.75 mL of water to form a homogeneous suspension.

## Synthesis of modified precursors with different silica shell thicknesses

The synthesis of different silica shell thicknesses was carried out following a modification of the procedure by Li et al.[69] In summary, the procedure involves the preparation of silica-coated particles using a 2.6 mg mL$^{-1}$ aka@PAA mass concentration.

$d_{SiO2} < 2.5\,nm$: Initially, the resuspended aka@PAA sample (1.75 mL) is mixed with 40 mL of absolute ethanol and treated with ultrasound for 30 minutes. Following this, 250 μL of ammonia solution and TEOS (amounts in Supplementary Table 10.3) are added dropwise to the dispersion, which is stirred overnight. The mixture is then centrifuged at 12,000 rpm for one hour at 20 °C, after which the precipitate is washed with 10 mL of ethanol and water and resuspended in 1 mL of water.

$d_{SiO2} > 3\,nm$: The initial dispersion of 1.75 mL aka@PAA in 40 mL ethanol is divided into two flasks. In each flask, 125 μL of ammonia solution is added while stirring. Different amounts of TEOS (see Supplementary Table 10.3) are introduced, with an additional portion of the same amount of TEOS added to the second flask after one hour if mentioned. Both mixtures are stirred overnight, centrifuged at 12,000 rpm for 15 minutes, washed with 15 mL ethanol once, and with 15 mL water twice. The resulting precipitates are resuspended in 1 mL of water.

## Synthesis of cobalt ferrite particles *via* hydrothermal reaction

For the hydrothermal synthesis, 0.302 g FeCl$_3 \cdot$6H$_2$O (97%, 1.12 mmol), 0.151 g FeCl$_2 \cdot$4H$_2$O (0.757 mmol), and 0.090 g CoCl$_2 \cdot$6H$_2$O (0.377 mmol) were dissolved in ultrapure water ($V_{DIW}$, see Supplementary Table 10.4) by stirring with a short cylindrical magnetic bar (length = 10 mm, diameter = 2 mm) in a 25 mL Teflon liner for 5 minutes. Subsequently, 1.13 mmol of the modified nanorod solution (0.1 g particle mass, respectively 4 mg mL$^{-1}$ nanorod mass concentration in resulting reaction solution) was added under stirring, followed by swiftly adding 12.5 mL of 1 M sodium hydroxide solution (NaOH) for a filling volume of 100% ($V = 25$ mL). All quantities were reduced equivalently when only 55% of the reactor was filled ($V = 13.25$ mL). Then, the hydrothermal reactor, fitted with the Teflon inlet, was sealed. The system was flushed with nitrogen for 30 seconds and then pressurized to a pressure of 10 bar. Following a 30-minute temperature ramp to 160 °C or 190 °C, the solution was stirred at 500 rpm for 23.5 hours at 160 °C. Afterward, the particles were magnetically separated and washed thoroughly three times with Milli-Q water. The exact weightings and parameters of the individual formulations are summarized in the Supplementary Table 10.4.

The citrate coating of the samples was performed according to the methodology established by Nappini et al.[151] Negatively charged particles were generated by dispersing and magnetically stirring them overnight in 15 mL of fresh 0.25 M tetramethylammonium hydroxide solution. These dispersions were then integrated into a 100 mM citric acid solution, resulting in a target weight percentage of around 0.25 wt.%. The resultant mixture was gently stirred for two hours at room temperature. Finally, the acquired particles were isolated *via* magnetic decantation and further stirred in an equal volume of 20 mM trisodium citrate solution for 24 h. The stabilized solutions underwent dialysis against milli-Q water for one week, followed by the determination of the mass fractions of the citrate-stabilized particle solutions using gravimetric analysis. The samples were adjusted to a concentration of 1 wt.% using an evaporator.

## Characterization

**Mass fractions.** Mass fractions of the particle solutions were determined by evaporating 0.5 mL of the suspension in a laboratory oven at 90 °C and

using the following Eq. 9:

$$\omega = \frac{m(\text{MNP, dry})}{m(\text{ferrofluid, liquid})} \cdot 100\% \qquad (9)$$

**Elemental analysis (EA).** The iron and cobalt content was analyzed with inductively coupled plasma-atomic emission spectroscopy (ARCOS from Fa. Spectro). Before the measurement, 500 μL of the 1 wt.% samples were digested with 3 mL of inverse aqua regia solution (HNO$_3$/HCl = 3/1) at 280 °C using a START-1500 T-280 microwave (Fa. MLS Leutkirch GmbH, Germany) and diluted to 20 mL. Each measurement was performed twice. The cobalt-to-iron ratio $\kappa$ was calculated as follows (Eq. 10), resulting in Supplementary Equation 4.2 and Supplementary Equation 4.3 for cobalt (x) and iron contents (3-x) for the formula Co$_x$Fe$_{3-x}$O$_4$, respectively:

$$\kappa = \frac{n(\text{Co})}{n(\text{Fe})} \qquad (10)$$

**Dynamic light scattering (DLS) and Zeta potential (ZP).** The hydrodynamic diameters ($d_H$) were determined using a ZetasizerPro Blue (Malvern Panalytical Ltd., Malvern, UK) with the software ZS Explorer 3.2.0.84 (Malvern Panalytical Ltd., Malvern, UK). A bubble-free injection of approximately 0.7 mL of a 0.01 wt.% suspension was filled into a disposable folded capillary cell (Product number: DTS1070, Malvern Panalytical, Germany), which was rinsed with ethanol and water prior. The hydrodynamic size was determined using a measuring angle of 173°. Each measurement was conducted thrice over 60 seconds, with a waiting time of 120 s, and the system temperature was maintained at 25 °C. The surface charge of the nanoparticles was determined based on three measurements using the same protocol.

**Fourier Transform Infrared (FTIR) Spectroscopy.** Approximately 1 mg of the dried MNP sample was finely ground with 2 – 3 spatula tips of potassium bromide (stored at 60 °C in a drying oven), and the resulting mixture was compressed at a pressure of 5 tons. The resultant pellet was analyzed with a resolution of 4 cm$^{-1}$ by averaging 128 scans using a Bruker FTIR Vertex 70 spectrometer (Bruker Optics GmbH & Co. KG, Ettlingen, Germany). Spectra were collected in transmission mode over a range of 4000 to 400 cm$^{-1}$. The data acquisition and analysis were performed using Opus 8.7 software (Bruker Optics GmbH & Co. KG, Ettlingen, Germany).

**High-Resolution Transmission Electron Microscopy (HR-TEM) with Energy Dispersive X-ray spectroscopy (EDX).** HR-TEM images were obtained using a double-corrected JEOL JEM 2200FS microscope (acceleration voltage: 200 kV, with Cescor and CETCOR Cs correctors from CEOS). The elemental composition and mapping of the nanoparticles were verified using an Oxford X-Max 100 TLE EDX detector. Copper grids, prepared as described for TEM measurements, were employed for sample mounting. To ensure a representative analysis, images were captured from at least two different areas on the grid, and additionally, by performing at least three line scans.

**Small-Angle X-ray Scattering (SAXS).** The SAXS data were taken at the beamline ID10@ESRF (see Supplementary Section S9, https://doi.org/10.15151/ESRF-DC-2160804078). The magnetic chamber from the ID02 beamline was inserted as a sample environment. This allowed the application of a field perpendicular to the beam to be varied between about 3 mT and 930 mT according to the gap between the magnets (see Supplementary Fig. 9.1). The energy was set to 7 keV, just below the iron edge. The Eiger4M was used as a detector. The sample detector distance was about 7.05 m. The nanoparticles were dispersed in an 80 wt.% glycerol/water mixture to increase stability, similar to the VSM measurements. The samples were inserted into quartz capillaries with a diameter

of 1.00 mm and a wall thickness of 0.01 mm. The capillaries were sealed with hot glue before the measurement. The 2D patterns of each magnetic field strength (see Supplementary Fig. 9.2a) were angularly averaged with the dynamix XPCS data analysis package provided by ID10@ESRF. These scattering images were subsequently masked in various ways to allow for evaluation averaged over all angles or for angles perpendicular or parallel to the magnetic field (see mask Supplementary Fig. 9.2b). The averaged intensities were analyzed using the Beaucage model Eq. 2 *via* a Matlab® (Version 2024a) code using the curve-fitting toolbox.

**Transmission Electron Microscopy (TEM).** The characterization of the size and morphology of the nanoparticles was conducted utilizing Transmission Electron Microscopy (TEM) on a FEI Tecnai G2 Spirit Twin, operated at an accelerating voltage of 120 kV. 200 mesh copper grids from Science Services GmbH (Munich, Germany) were used for sample preparation. These grids were coated with carbon films, which were synthesized through a vapor deposition process developed at the University of Hamburg. A 10 µL aliquot of a 1 wt.% particle dispersion, diluted with 1 mL of ultrapure water, was applied to the grid, followed by air drying at room temperature. TEM imaging was performed across at least five distinct grid regions to ensure an adequate representation of particle size distribution. Particle diameters were assessed by measuring 150 particles from each sample, employing the open-source software ImageJ (62, Version 1.50i, National Institutes of Health, USA). The first moment of the particle size distribution, denoted as $d_{\text{TEM}}$, was extracted from the log-normal distribution function, with the corresponding median ($\mu$) and variance ($\sigma^2$) expressed in Eq. 11 and resulting dispersity values in Supplementary Equation 4.4.

$$d_{\text{TEM}} = \exp\left(\mu + \frac{\sigma^2}{2}\right) \tag{11}$$

**Vibrating-Sample Magnetometer (VSM).** These measurements were conducted with an EZ-9 magnetometer (MicroSense). The samples were measured using the following protocol of a symmetric full-loop recipe with an applied field strength from -2500 to -1000 mT with step size of 500 mT, −1000 to -500 mT with a step size of 100 mT, -500 to 500 mT with a step size of 25 mT, 500 to 1000 mT with a step size of 100 mT, 1000 to 25,000 mT with a step size of 500 mT, and backyards. Around 70 µL of 1.0 wt.% citrate-stabilized particle suspensions (aqueous and 80 wt.% glycerol/water mixture) were pipetted in a 6 mm ULTEM cup, which was affixed to an 8 mm quartz glass sample holder using double-sided tape and Parafilm. Background measurements were carried out with Milli-Q water or 80 wt.% glycerol/water mixture. Data acquisition was facilitated through the software EasyVSM 20180925 (MicroSense). The accuracy of the magnetic field measurement at the 3.5 mm sample gap is specified as 1% of the reading or ±0.05% of the full scale. The measurement data were normalized to the mass of the nanoparticles obtained *via* gravimetry. The aqueous samples were measured twice, whereas the glycerol-based suspensions were measured three times.

**X-ray diffraction (XRD).** XRD patterns were acquired using a Philips X'Pert PRO MPD diffractometer (Almelo, The Netherlands), utilizing an X-ray wavelength ($\lambda$) of 154.06 pm. Data were collected across a 2θ range from 10° to 100° employing Cu Kα radiation. 100 µL aliquot of the stabilized aqueous particle dispersion was applied to a silicon wafer with an orientation of (911) ± 0.5° and subsequently allowed to air dry for sample preparation. All patterns were background-adjusted and normalized to [0, 1] depending on the maximum intensity. The evaluation of the crystal phases was done with the software Highscore X'pert PRO by PanAnalytical (Version 2.2.3, Almelo, The Netherlands) with the Joint Committee on Powder Diffraction Standards (JCPDS) PDF no. 00-034-1266 for akaganeite, PDF no. 00-003-0864 for cobalt ferrite, and PDF no. 00-046-1045 for silicium dioxide as well as PDF no. 01-072-1668 for halite (sodium chloride).

## Data availability
The datasets generated during and/or analyzed during the current study are available in the Supplementary Information, under the ESRF link given in the Method Section, or from the corresponding author on reasonable request if they are not depicted in this work or the SI.

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

## Acknowledgements
This work was funded by the Deutsche Forschungsgemeinschaft (DFG) within the project "RTG 2536." We acknowledge the European Synchrotron Radiation Facility (ESRF) for the provision of synchrotron radiation facilities under proposal number SC 5638, and we would like to thank Yuri Chushkin for assistance and support in using beamline ID10. We further gratefully acknowledge Marthe Kaufholz, Felix Lehmkühler, and Robert Bauer for their intensive help during the experiment. We gratefully acknowledge the ESRF for providing and installing the magnetic sample chamber from ID02. They would like to express special thanks to Stefan Werner for conducting the TEM measurements as well as Andrea Köppen for performing the HR-TEM and EDX measurements. The authors acknowledge Charis Schlundt and Nina Schober, who conducted all of the XRD experiments. The authors also thank the team at the Element Analysis Center of the University of Hamburg for performing the F-AAS measurements. We acknowledge the financial support from the Open Access Publication Fund of Universität Hamburg.

## Author contributions
M.W.: Conceptualization, Investigation, Methodology, Validation, Writing – original draft, review & editing. J. K.: Investigation, Methodology – original draft. B.H.: Conceptualization, Methodology, Validation, Writing – review & editing, Supervision, Funding acquisition. All authors reviewed and approved the submission of the manuscript.

## Funding

## Competing interests
The authors declare no competing interests.
