## [Transparent Peer Review file · Communications Chemistry]

Magneto-responsive Chain-like Arrangements of Size-tuned and Cobalt-doped Ferrites derived from Silica-encapsulated Precursors

Corresponding Author: Ms Maria Weißpflog

Version 0:

Reviewer comments:

Reviewer #1

(Remarks to the Author)

Comments:

The manuscript presents interesting work on “Magneto-responsive Chain-like Arrangements of Size-tuned and Cobalt-doped Ferrites derived from Silica-encapsulated Precursors”. While the study has potential scientific merit, several key aspects of characterization and interpretation require clarification and further experimental support to substantiate the claims. Therefore, I recommend a major revision. The authors should address all the above points with additional experiments, thorough data analysis, and more precise explanations to substantiate their claims.

1. Abstract: There is no direct quantification of the elemental composition. It is strongly recommended that inductively coupled plasma optical emission spectroscopy (ICP-OES) be conducted to provide accurate and quantitative analysis of the constituent elements, thereby supporting the compositional claims.

2. Pg. 3, L 40: The assembly can be achieved by various methods, e.g., a spontaneous self-assembly, which appears incomplete and lacks specificity. The authors are encouraged to elaborate on other applicable methods to support this statement.

3. Pg. 7, L 111: The role of Na₂HPO₄ in promoting anisotropic growth via interaction with the (001) plane requires further mechanistic explanation. A discussion supported by relevant literature or theoretical insight into how Na₂HPO₄ preferentially interacts with specific crystallographic facets would help clarify the surface dynamics underlying nanorod formation.

4. Pg. 7, L 115-117: While the diffraction data is well-matched with JCPDS PDF No. 00-034-1266, identifying the material as tetragonal akaganeite, the discussion on anisotropic growth remains superficial. The authors should consider elaborating on the specific facet orientations and associated intensity variations observed in the XRD pattern, as these are critical in understanding the directional growth and morphological evolution of the nanostructures.

5. Pg. 8, L 129-130: The statement regarding surface charge ($\zeta = -39.0 \pm 0.6$ mV) merits further scrutiny. Zeta potentials beyond ± 30 mV typically indicate colloidal stability, yet the authors mention possible agglomeration. Could the observed agglomeration be attributed to increased magnetic interactions or dipole–dipole assembly of magnetic nanoparticles leading to a thermodynamically favorable configuration? A brief discussion of this phenomenon would enrich the interpretation.

6. Pg. 10, Pg 191: The manuscript refers to the shell thickness of the nanostructures to be 2.5 nm. However, no high-resolution transmission electron microscopy (HRTEM) images are presented to support this claim. The authors are requested to clarify how the shell thickness was determined. Provide HRTEM or line-scan profiles from elemental mapping, as it is difficult to assess the accuracy of the dimensions.

7. Pg. 16, L 294: The statement discusses that when the silica shell thickness is 4 nm, the layer is sufficiently thick to become impermeable. But no characterization of silica shell density is conducted to prove its impermeability. I recommend performing characterization techniques such as BET and HRTEM to determine porosity, or citing recent studies to support this claim.

8. Pg. 19, L 371: The statement mentions that at high magnetic field strength, chain elongation takes place as there is an increase in nanochain alignment and thus an increase in magnetic moment, but the correlation between the length of the

nanochain and magnetic hysteresis results, like remanence or coercivity, is not quantified. Kindly clarify this, as magnetic hysteresis results are susceptible to chain length.

Reviewer #2

(Remarks to the Author)

The manuscript by Weißpflog et al. reports on an eco-friendly aqueous synthesis of cobalt-doped ferrite $\text{Co}_x\text{Fe}_{3-x}\text{O}_4$ nanoparticles. Authors used β -FeOOH nanorods, surface-modified with sodium polyacrylate and a silica layer of varying thickness, as precursors for a hydrothermal conversion process. The authors demonstrate that this method allows for control over nanoparticle size (14–47 nm) and stoichiometry (Co/Fe ratio) by tuning the reaction temperature and the thickness of the initial silica shell. Authors used a lot of advanced experimental techniques to support their findings. I find this manuscript well structured, very comprehensive, interesting to read, and worth publishing in this journal. However, I would kindly ask to address the following comments:

1. The authors should explicitly state what makes this method superior to or fundamentally different from existing methods for synthesizing ferrite nanochains.
2. I missed somehow comparison of the magnetization/hysteresis behaviour of the proposed nanoparticles with the “gold standard” of magnetic nanoparticles Fe_3O_4 .
3. Authors mentioned several times potential of biomedical applications of the proposed nanoparticles, e.g., magnetic hyperthermia. Can authors comment on biodegradability of nanoparticles and heating abilities of these nanoparticles compared to well studied nanoheaters Fe_3O_4 . Generally, the authors should be more specific about the potential applications. For example, if for biomedicine, are the magnetization values and particle sizes suitable for MRI contrast, hyperthermia, or others?

Reviewer #3

(Remarks to the Author)

The manuscript presents a detailed study on the synthesis and characterization of cobalt-doped ferrite nanoparticles arranged in chain-like structures, derived from silica-encapsulated akaganeite precursors. The authors explore the influence of precursor modifications (polyacrylic acid (PAA) and silica coatings), synthesis conditions (temperature, silica shell thickness), and cobalt doping on particle size, composition, and magnetic properties. The work is supported by comprehensive experimental data, including TEM, XRD, FTIR, AES, VSM, and SAXS measurements, provided in both the main text (MS) and Supporting Information (SI).

The study is scientifically sound, with a clear focus on magneto-responsive nanomaterials, which aligns well with the scope of Communications Chemistry. The approach of using silica-encapsulated precursors to control particle size and arrangement is innovative, and the detailed characterization provides valuable insights into structure-property relationships. The SI is exceptionally detailed, covering synthesis protocols (S1–S3, S10), characterization data (S4–S6, S8–S9), and mechanistic insights (S7). Techniques such as TEM, XRD (Debye-Scherrer and Williamson-Hall analyses), AES, FTIR, VSM, and SAXS are appropriately used to characterize the materials.

The manuscript presents a compelling and original study with significant potential for Communications Chemistry. However, organizational issues, incomplete mechanistic discussions, and limited application insights should be addressed to meet the journal's standards. I recommend the authors undertake revisions, focusing on the points outlined below, and resubmit for further review. With these improvements, the work could make a strong contribution to the field of magnetic nanomaterials.

- The main text MS lacks a clear structure, with some sections blending experimental details and interpretations in a way that obscures key findings.
- The introduction does not sufficiently contextualize the work relative to recent advances in cobalt ferrite synthesis, citing older references without discussing state-of-the-art methods like microwave-assisted synthesis or greener approaches.
- Some figure captions are overly descriptive and lack conciseness, making it difficult to quickly grasp the key points.
- Section S7 on the mechanism of cobalt ferrite chain formation is truncated, limiting the understanding of how silica shells and PAA influence chain assembly. The MS should expand on this, possibly integrating a schematic diagram to illustrate the proposed mechanism.
- The effect of temperature on cobalt ferrite phase purity is described, but the underlying chemical or thermodynamic factors are not addressed.
- The SI reports large standard deviations for some measurements (e.g., $\sigma_{\text{TEM}} = 10.8$ nm for CF-S3-1, SI, p. 6), indicating high polydispersity, but the MS does not discuss implications for practical applications.
- Some TEM images show empty silica shells which gives the impression of inhomogeneous and uncontrolled silanization of the SiO_2 -coated aka@PAA . Therefore, it would be great to see EDX mapping to properly see the localization of akaganeite in the SiO_2 -coated aka@PAA . Also, EDX mapping of CF nanoparticles to show the localization of Co and Fe is recommended.
- The MS focuses heavily on synthesis and characterization but does not adequately discuss how the chain-like arrangements enhance magneto-responsive properties compared to isotropic nanoparticles. For example, how do the chains improve magnetic anisotropy or coercivity?
- Potential applications are mentioned briefly in the introduction but not revisited in the discussion. A section on how the findings could impact fields like magnetic hyperthermia or other field would strengthen the manuscript.
- The MS and SI contain minor typographical errors, such as "akaganeite" instead of "akaganeite".

Reviewer comments:

Reviewer #1

(Remarks to the Author)

The authors have satisfactorily addressed all the comments and queries raised during the review process. The revised manuscript incorporates the suggested changes, and the quality of the paper has improved significantly.

I have found no further major concerns, and the manuscript is now suitable for publication in its present form. Therefore, I recommend its acceptance.

Reviewer #2

(Remarks to the Author)

Comments were properly addressed. I recommend this manuscript for publication.

Reviewer #3

(Remarks to the Author)

The authors have satisfactorily addressed the initial comments and substantially improved the manuscript. I recommend the article for publication in its current form.

Dear Editor, dear Reviewers,

We thank the reviewers for their valuable feedback and for evaluating our article for acceptance following a comprehensive revision and a review of the revised version. We have substantially revised the main manuscript and the supplementary information (SI), incorporating essential information in response to the reviewers' comments. All modifications in the manuscript and SI are highlighted in red ("highlighted Version"). Below, we present a detailed response addressing each comment.

Reviewer #1 (Remarks to the Author):

The manuscript presents interesting work on "Magneto-responsive Chain-like Arrangements of Size-tuned and Cobalt-doped Ferrites derived from Silica-encapsulated Precursors". While the study has potential scientific merit, several key aspects of characterization and interpretation require clarification and further experimental support to substantiate the claims. Therefore, I recommend a major revision. The authors should address all the above points with additional experiments, thorough data analysis, and more precise explanations to substantiate their claims.

We thank Reviewer #1 for the valuable feedback and constructive suggestions. We have made substantial revisions to the characterization of the precursor system, particularly regarding its composition and structure, in order to clarify and better support the claims presented in the manuscript. Through these modifications, we aim to substantiate our key findings and enhance the overall scientific rigor of our work. Once again, we sincerely appreciate the helpful comments, which have greatly contributed to enhancing the quality of our manuscript.

1. Abstract: There is no direct quantification of the elemental composition. It is strongly recommended that inductively coupled plasma optical emission spectroscopy (ICP-OES) be conducted to provide accurate and quantitative analysis of the constituent elements, thereby supporting the compositional claims.

In the original manuscript, we performed F-AAS measurements instead of ICP-OES measurements on the cobalt-doped particle systems (with respect to the content of cobalt and iron). The corresponding data, as well as the equations, are presented in Section S4 of the supplementary material. However, due to the journal's 200-word limit for the abstract (which we adhered to), we were unable to specify the methodology in detail within that section. Regarding the reported range of 0.40 to 0.51, this pertains to all synthesized particles discussed in this work. We hope this clarification is helpful, but please let us know if we have misunderstood your comment.

We refer to pg. 22, line 414 and Supplementary Section S4. To confirm the Co/Fe ratio, line scans and EDX mappings were additionally performed on three samples, namely CF-A3 (Fig. 4d), CF-P3-1 (Fig. 4e), and CF-S3-8 (Fig. 4f). The CF samples show the same trend as in the F-AAS measurements, namely increasing Co/Fe ratio with PAA addition and further enhancement with increasing particle diameter. However, it can be observed that the absolute μ values of 0.205, 0.213, and 0.222 observed from the map sum spectra are higher than those determined by AAS. This discrepancy is due to the partial overlap of the Co and Fe $K\alpha_1$ lines, given inherent uncertainties in the intensity values. Additionally, differences in the average values observed through the line sum spectra and the map sum spectra can be identified (see Supplementary Fig. S6.5 – S6.10). The mapping, which analyzes above a larger number of particles, yields a value that approaches the AAS value (according to the significantly larger particle number). Furthermore, the atom percent distribution of the line scans enables an interpretation of the silicon presence. Interestingly, in the case of CF-S3-8, the silicon intensity at the

particle edges increases, which can be attributed to a local increase in silicon concentration (see Supplementary Fig. S6.9). This indicates the presence of a thin silica shell surrounding the particles, which stabilizes them within the chains or at least influences their formation mechanism.

2. Pg. 3, L 40: The assembly can be achieved by various methods, e.g., a spontaneous self-assembly, which appears incomplete and lacks specificity. The authors are encouraged to elaborate on other applicable methods to support this statement.

We incorporated relevant literature into this section to expand the discussion on self-assembly processes, which utilize molecular interactions/or external fields, and/or shape and entropy effects on pg. 3 lines 46 – 69.

3. Pg. 7, L 111: The role of Na₂HPO₄ in promoting anisotropic growth via interaction with the (001) plane requires further mechanistic explanation. A discussion supported by relevant literature or theoretical insight into how Na₂HPO₄ preferentially interacts with specific crystallographic facets would help clarify the surface dynamics underlying nanorod formation.

We incorporated a discussion of the anisotropic growth of akaganeite in the manuscript on pg. 9, lines 157. Therefore, we additionally performed HR-TEM, EDX, and line-profile scan measurements to present the crystal structure of the nanorods.

The role of Na₂HPO₄ in preventing akaganeite from transferring to hematite and promoting anisotropic growth due to its interaction with akaganeite is included in the SI in Section S1.2 with a brief reference in the main manuscript on pg. 9, line 168. We also included an illustration of the proposed mechanism in Fig. S1.1f. The anisotropic growth is promoted not only by the double chain orientation of the akaganeite structure but also by the presence of phosphate ions on the (hk0) faces of the initially synthesized akaganeite seeds. In particular, phosphate ions serve to prevent oxidation to hematite by hindering the agglomeration of various nanorods along the [010] and [100] directions.

4. Pg. 7, L 115-117: While the diffraction data is well-matched with JCPDS PDF No. 00-034-1266, identifying the material as tetragonal akaganeite, the discussion on anisotropic growth remains superficial. The authors should consider elaborating on the specific facet orientations and associated intensity variations observed in the XRD pattern, as these are critical in understanding the directional growth and morphological evolution of the nanostructures.

We included the facets using Miller indices in the diffractograms to visualize the crystal planes (Fig. S1.3) and expanded the discussion on anisotropic growth and preferentially exposed faces in Section S1.2 and Section S1.3. Therefore, we performed an additional XRD experiment (Fig. S1.3b) to mitigate orientational effects that can arise from drop-casting the dense suspension on a silicon wafer. We only included a brief reference in the manuscript on pg. 9, line 172, to a more detailed discussion in Section S1.3, as the precursor is only used as an iron source for the synthesis of the cobalt-doped particles, which represents the main focus of the work. Additionally, HR TEM images and SAED patterns were analyzed.

5. Pg. 8, L 129-130: The statement regarding surface charge ($\zeta = -39.0 \pm 0.6$ mV) merits further scrutiny. Zeta potentials beyond ± 30 mV typically indicate colloidal stability, yet the authors mention possible agglomeration. Could the observed agglomeration be attributed to increased magnetic interactions or dipole–dipole assembly of magnetic nanoparticles leading to a thermodynamically favorable configuration? A brief discussion of this phenomenon would enrich the interpretation.

We have included a cautious interpretation of the relationship between zeta potential values and the potential agglomeration of particles as a function of pH on pg. 11, line 199. We refer to the work of Lowry *et al.*, where the historically accepted value of ± 30 mV for moderately stable suspensions is

critically evaluated, as this value was determined for micro-sized blood cells. As the particle diameter decreases, Van der Waals forces become more dominant, increasing the likelihood of particle agglomeration at the same zeta potential values. Since the akaganeite particles are antiferromagnetic, we also assume that there are no significant magnetic interactions (unless Van der Waals forces are included in that consideration).

By introducing a PAA shell, the zeta potential further increases in magnitude, and d_H as well. If steric effects are introduced through a 'protective' polymer shell, lower surface charges may be adequate, because the steric barrier hinders particle interactions. This may also indicate that the fraction with $d_H \approx 220$ nm corresponds to individually PAA-stabilized particles. However, due to the significantly increased agglomeration observed during TEM preparation in comparison to bare akaganeite (at the pH of the PAA modification), this explanation is rather unlikely. We included this discussion on pg. 12, line 216.

6. Pg. 10, Pg 191: The manuscript refers to the shell thickness of the nanostructures to be 2.5 nm. However, no high-resolution transmission electron microscopy (HRTEM) images are presented to support this claim. The authors are requested to clarify how the shell thickness was determined. Provide HRTEM or line-scan profiles from elemental mapping, as it is difficult to assess the accuracy of the dimensions.

The calculation of the silica layer thickness was performed using Eq. S3.1, assuming that the diameter of the embedded akaganeite rods corresponds to the mean value obtained from Supplementary Sect. S1. The analysis of each silica modification was demonstrated in Sect. S3. We have implemented Fig. S3.1 to visualize the layer thickness calculation by an illustration. We implemented the histograms with lognormal distribution curves for these fractions in Sect. S3 (Fig. S3.3) for a better understanding of the analysis.

Additionally, we performed HR-TEM imaging combined with EDX analysis and line scans on two synthesized Aka@0.12PAA@SiO₂ samples and revised Fig. 2 accordingly. To demonstrate the presence of akaganeite in the core of the silica precursors and to evaluate the porosity of the silica layers, two fractions were analyzed with $d_{SiO_2} < 4$ nm and $d_{SiO_2} > 4$ nm (see Supplementary Sect. S3.2). In the HR-TEM images, a clear distinction between akaganeite and silica can be observed (Fig. 2h and Fig. 2j, see Supplementary Fig. S3.4). Additionally, Energy Dispersive X-ray spectroscopy (EDX) analysis reveals silicon signals at the surface and edges of the particle, whereas iron is present primarily in the core, especially for the sample with the thicker shell. The transition is clearly visible in the line-scan profiles (Fig. 2k, see Supplementary Fig. S3.6). However, in the sample with the thin silica shell, where the line scans do not appear to be very conclusive (Fig. 2i, see Supplementary Fig. S3.5), a Si/Fe ratio of 0.28 is observed based on the distribution in the sum spectra (see Supplementary Fig. S3.7). This is significantly different from the Si/Fe ratio of 0.09 that was determined in the bare akaganeite sample (see Supplementary Fig. S1.2). In the sample with a thicker shell, the silicon content correspondingly increases, resulting in a Si/Fe ratio of 2.7 (see Supplementary Fig. S3.7). Chloride atoms, located within the tunnel structures of akaganeite, can also be detected. This confirms that akaganeite remains stable and is neither dissolved nor transformed into another phase during the PAA functionalization and the Stöber process. We implemented this discussion on pg. 14, line 265.

7. Pg. 16, L 294: The statement discusses that when the silica shell thickness is 4 nm, the layer is sufficiently thick to become impermeable. But no characterization of silica shell density is conducted to prove its impermeability. I recommend performing characterization techniques such as BET and HRTEM to determine porosity, or citing recent studies to support this claim.

In summary, we have revised our wording regarding porosity, as we could not provide meaningful evidence of this parameter, relating the assessment of akaganeite to the thickness of the silica layer over time. Additionally, the HR-TEM analysis allows for a cautious interpretation of the porosity of the silica shells, which lack crystal planes and are therefore more likely to exhibit an amorphous or microporous (pore diameter of <2 nm) structure. Comparable porosities have been investigated in the literature, as explained later, and determined using BET analysis for similarly synthesized silica shells produced *via* the Stöber process. Such a structure could dissolve under alkaline conditions, with the dissolution time depending on the thickness of the silica layer. We refer to the corresponding literature on pg. 15, line 279 and discuss the porosity in Supplementary Section S3.2.

8. Pg. 19, L 371: The statement mentions that at high magnetic field strength, chain elongation takes place as there is an increase in nanochain alignment and thus an increase in magnetic moment, but the correlation between the length of the nanochain and magnetic hysteresis results, like remanence or coercivity, is not quantified. Kindly clarify this, as magnetic hysteresis results are susceptible to chain length.

We thank Reviewer #1 for the valuable suggestions. Please note that the VSM measurements were performed on suspensions in water and in 80% glycerin/water mixtures, and not on solids. There, the moments can align relatively easily along the external field in the presence of the external field. The measurements in glycerin do not fully describe immobilized particle systems. In this case, particles can align much more easily than in solids, and the measured coercivities should therefore be regarded less as an intrinsic material property of single-domain or pseudo-single-domain particles, assuming a non-interacting system. As we are additionally dealing with large aggregates and not with a "pure" suspension or even a dispersion, the dipole-dipole interaction within the existing aggregates could lead to an effective crystal anisotropy, thereby changing the magnetic behavior. Based on these points, we primarily discussed chain formation in relation to the size ratios derived from TEM and VSM (see Fig. 9). We briefly discuss the influence of the saturation magnetization, squareness ratio and coercivity on pg. 47, line 912 and refer to the Supplementary Fig. S9.12 to S9.14. Additionally, we have included a comprehensive section for a cautious classification and description of the particle systems, focusing on their SAR and potential for biomedical applications (last section of the result and discussion part). In this context, the coercivities, saturation magnetizations, and remanence values determined in Fig. 5 were also considered in this discussion on pg. 50/51, line 984 – 1021.

Reviewer #2 (Remarks to the Author):

The manuscript by Weißpflog et al. reports on an eco-friendly aqueous synthesis of cobalt-doped ferrite $\text{Co}_x\text{Fe}_{3-x}\text{O}_4$ nanoparticles. Authors used β -FeOOH nanorods, surface-modified with sodium polyacrylate and a silica layer of varying thickness, as precursors for a hydrothermal conversion process. The authors demonstrate that this method allows for control over nanoparticle size (14–47 nm) and stoichiometry (Co/Fe ratio) by tuning the reaction temperature and the thickness of the initial silica shell. Authors used a lot of advanced experimental techniques to support their findings. I find this manuscript well structured, very comprehensive, interesting to read, and worth publishing in this journal. However, I would kindly ask to address the following comments:

We thank Reviewer #2 for the positive feedback and for considering our paper as publishable with only minor revisions. We are delighted to hear that our manuscript represents a significant contribution to Comms. Chem. and appreciate reviewer #2's recognition of the intensive work compared with the design of magnetic nanoparticles shown in this manuscript. We promptly addressed the suggestions for minor changes to further improve the quality of our work.

1. The authors should explicitly state what makes this method superior to or fundamentally different from existing methods for synthesizing ferrite nanochains.

We included a more precise explanation of the choice of nanochain arrangement in the introductory section on pg. 3. Bioinspired magnetic nanochains like magnetosomes served as a model for this work, as they exhibit very promising magnetization and are discussed concerning cancer therapies and their potential for medical use. Doping with cobalt ions has also been investigated to study the effects of increased magnetic anisotropy. However, it should be noted that their application remains controversial, primarily because of the limited scalability of cultivation and purification. Nonetheless, this research area is highly prominent, and synthetic strategies are being developed to replicate these systems and facilitate their practical application. Considering the synthesized particle morphology in our work, magnetosome-like structures are primarily observed in larger individual particle diameters.

Using silica-encased precursor particles can result in the formation of nanochain-like structures, since the silica shell influences the formation mechanism. We have now explained this mechanism more explicitly in an additional section on pg. 24 (Fig. 4) to highlight the fundamental differences compared to other methods.

Due to word count limitations in the introduction part, we could only briefly describe the differences from other existing synthesis methods or approaches in the last section of the introduction. Since we have included a section on potential applications and the capabilities of the particle systems on pg. 4, line 64 and pg. 5, line 84, the key highlights will be further emphasized.

1. Essentially, through systematic investigations of precursor-derived synthesis with a hydrothermal step in previous studies, we have determined that cubic, truncated octahedral, coffin-like, and spherical particles can be produced *via* akaganeite in combination with metal salts. Reducing the proportion of cobalt salt additionally enables the synthesis of cobalt-doped ferrite particles, which we performed within these morphologies. We would like to reference the systematic work of Flores Urquizo *et al.* (10.1002/admi.202300206), who have demonstrated that bare cobalt-doped particles, especially in the range of x from 0.2 to 0.6, exhibit lower toxicity compared to iron oxides ($x = 0$) and cobalt ferrite ($x = 1$). The coating of the particles with 3-(2-aminoethyl amino) propyltrimethoxysilane (AEPTMS) improved the system's biocompatibility and stability without significantly affecting cytotoxicity, and it even

increased cell viability. They assume that the presence of the coating likely enhanced the nanoparticles' stability and resistance to degradation. Thus, one goal of our work was to achieve such a cobalt doping, which we successfully did with a cobalt-to-iron ratio of 0.2, respectively $x \approx 0.5$.

2. In addition to the toxicity evaluations, it has already been shown in literature and also in our group that the incorporation of cobalt ions leads to an increased magnetocrystalline anisotropy of the particles compared to iron oxide. It is observed that an increase in the anisotropy constant (which consists of magnetocrystalline contributions) leads to a reduction in the average particle size that produces the highest hyperthermia performance. (<https://doi.org/10.1063/1.2830975>) As a result, cobalt ferrite particles should already be dominated by the Brownian relaxation mechanism in the high single-digit nanometer range (around 7 nm), compared to magnetite nanoparticles (around 14 nm). For example, an increase in the heating efficiency can be achieved by doping magnetite with cobalt to produce $\text{Co}_x\text{Fe}_{3-x}\text{O}_4$ within the concentration range of $x \leq 0.5$. (<https://doi.org/10.1016/j.physb.2020.412429>) Considering the different relaxation processes, the potential of the cobalt-doped systems is quite significant.
3. Additionally, realizing the silica-precursor-based synthesis could allow for further upscaling in hydrothermal reactors, as shown for precursor-derived hydrothermal methods (10.1039/D5RA02233A) by simultaneously enhancing the magnetic anisotropy, considering the future potential of this approach.

We further expand the conclusion section on pg. 55, line 1102 and pg. 56, line 1138.

2. I missed somehow comparison of the magnetization/hysteresis behaviour of the proposed nanoparticles with the "gold standard" of magnetic nanoparticles Fe₃O₄.

Since we are working in a range between cobalt ferrite and magnetite, and the advantages of cobalt ferrite over magnetite are highlighted in the introduction, we discussed the magnetism section (and mostly all other sections) in comparison to cobalt ferrite with $x = 1$. However, for example, we also included saturation magnetization values of magnetite and referred to this, especially in series 3 (comprising particles produced with increasing silica thickness). The higher iron content in cobalt-doped ferrites increases the saturation magnetization values, which can reach up to 100 Am²/kg for bulk magnetite. This is also observed in magnetite-like particles with low cobalt content in the CF-A samples (see Supplementary Fig. S8.1c).

3. Authors mentioned several times potential of biomedical applications of the proposed nanoparticles, e.g., magnetic hyperthermia. Can authors comment on biodegradability of nanoparticles and heating abilities of these nanoparticles compared to well-studied nanoheaters Fe₃O₄. Generally, the authors should be more specific about the potential applications. For example, if for biomedicine, are the magnetization values and particle sizes suitable for MRI contrast, hyperthermia, or others?

As described in comment #1, the synthesis of nanochain structures represents an intensive area of research. Various applications arise from the enhanced magnetic and shape anisotropy, the reversibility through external fields, the increased (rough) surfaces, magnetic coupling effects, and other properties, as explained below. We refer to this in more detail in the introduction. We want to highlight the work of various groups that synthesize silica-coated magnetic particle systems (primarily using Fe₃O₄, but also Co-Pd particles) with diverse applications, including, e.g., scaffolds to accelerate bone regeneration (<https://doi.org/10.1186/s40824-022-00278-2>), planar design of microfluidic biochips (10.1038/s41467-018-04172-1), catalysis (<https://doi.org/10.1039/D4NR02643H>), bacterial inhibition (<https://doi.org/10.1002/adv.202309564>), Osteoclast-targeted inhibition (with loaded

zoledronate, a medication for treating bone diseases) and heterogeneous nanocatalysis (10.1002/adma.201707515), and enzyme support and nanostirrer in biocatalysis (<https://doi.org/10.1021/acsami.0c03220>). We want to note that, due to the limited volume of the prepared samples, we were unable to perform any hyperthermia or toxicity measurements on our samples. This would, of course, also be a fascinating study, but is not the focus of this study.

However, we have included a comprehensive section for a cautious classification and description of the particle systems, focusing on their SAR and potential for biomedical applications on pg. 48 – 55. In this context, the coercivities, saturation magnetizations, and remanence values determined in Fig. 5 were also considered in this discussion, along with their size and size distribution.

In the context of toxicity, cellular distribution, and biodegradation of cobalt ferrite and cobalt-doped ferrite nanoparticles, a link was established between the literature and our particles regarding their size, magnetization, and chain arrangements. Despite the controversial discussion, it is worth noting that cobalt ferrite particles of smaller sizes can achieve comparable SAR values to those of magnetite. These smaller particles are expected to evade the immune system more effectively and have a longer circulation time in blood vessels, thereby increasing the likelihood of reaching the tumor tissue. Additionally, for magnetic carriers, small nanoparticles are more stable against aggregation, which prevents precipitation and consequently reduces the risk of blood vessel occlusion. This combination of high SAR performance and enhanced biological stability underscores the importance of nanoscale cobalt ferrite particles for biomedical applications. In conclusion, several studies have discussed the toxicity of cobalt ferrite and cobalt-doped ferrite particles, which are found to be size- and concentration-dependent. Interestingly, some refer to toxicity as a side effect and a factor to consider, with biocompatibility being enhanced through silica or polymer coatings. On the other hand, some approaches utilize the minimal release of Co (and Fe) for concurrent chemotherapy, since thermotherapies alone in most cases do not result in significant tumor reduction.

Reviewer #3 (Remarks to the Author):

The manuscript presents a detailed study on the synthesis and characterization of cobalt-doped ferrite nanoparticles arranged in chain-like structures, derived from silica-encapsulated akaganeite precursors. The authors explore the influence of precursor modifications (polyacrylic acid (PAA) and silica coatings), synthesis conditions (temperature, silica shell thickness), and cobalt doping on particle size, composition, and magnetic properties. The work is supported by comprehensive experimental data, including TEM, XRD, FTIR, AES, VSM, and SAXS measurements, provided in both the main text (MS) and Supporting Information (SI).

The study is scientifically sound, with a clear focus on magneto-responsive nanomaterials, which aligns well with the scope of Communications Chemistry. The approach of using silica-encapsulated precursors to control particle size and arrangement is innovative, and the detailed characterization provides valuable insights into structure-property relationships. The SI is exceptionally detailed, covering synthesis protocols (S1–S3, S10), characterization data (S4–S6, S8–S9), and mechanistic insights (S7). Techniques such as TEM, XRD (Debye-Scherrer and Williamson-Hall analyses), AES, FTIR, VSM, and SAXS are appropriately used to characterize the materials.

The manuscript presents a compelling and original study with significant potential for Communications Chemistry. However, organizational issues, incomplete mechanistic discussions, and limited application insights should be addressed to meet the journal's standards. I recommend the authors undertake revisions, focusing on the points outlined below, and resubmit for further review. With these improvements, the work could make a strong contribution to the field of magnetic nanomaterials.

We thank Reviewer #3 for the positive feedback and for considering our revised paper for further review. We appreciate reviewer #3's recognition of the detailed work shown in this manuscript and the SI. We are delighted to hear that our manuscript could represent a strong contribution to Comms. Chem. by considering the comments below. We promptly addressed the suggestions to improve the quality of our work.

- The main text MS lacks a clear structure, with some sections blending experimental details and interpretations in a way that obscures key findings.

We revised various sections in the manuscript to separate the experimental details from the interpretations. Usually, we presented the data first and interpreted them afterwards. Due to the numerous parameters and their associated effects, we added subscriptions to organize the script into a more transparent structure. Unfortunately, we were limited by the number of words in the introduction section, so we cannot include a brief description of the various sections. However, we incorporated this information in the first section of the results and discussion part and included a new Fig. 1 to improve the structure.

- The introduction does not sufficiently contextualize the work relative to recent advances in cobalt ferrite synthesis, citing older references without discussing state-of-the-art methods like microwave-assisted synthesis or greener approaches.

Due to word count limitations in the introduction section, we could only briefly describe the differences from other existing synthesis methods or approaches on pg. 5, lines 96 – 109. Since we have included a section on potential applications and the capabilities of the cobalt-doped particle systems on pg. 5, lines 85 – 93, the key highlights will be further emphasized. We have also included

significantly more recent literature. Still, we continue to reference the older references as well, as they describe essential initial steps in the development of this research field.

1. Essentially, through systematic investigations of precursor-derived synthesis with a hydrothermal step in previous studies, we have determined that cubic, truncated octahedral, coffin-like, and spherical particles can be produced *via* akaganeite in combination with metal salts. Reducing the proportion of cobalt salt additionally enables the synthesis of cobalt-doped ferrite particles, which we performed within these morphologies. We would like to reference the systematic work of Flores Urquizo *et al.* (10.1002/admi.202300206), who have demonstrated that bare cobalt-doped particles, especially in the range of x from 0.2 to 0.6, exhibit lower toxicity compared to iron oxides ($x = 0$) and cobalt ferrite ($x = 1$). The coating of the particles with 3-(2-aminoethyl amino) propyltrimethoxysilane (AEPTMS) improved the system's biocompatibility and stability without significantly affecting cytotoxicity, and it even increased cell viability. They assume that the presence of the coating likely enhanced the nanoparticles' stability and resistance to degradation. Thus, one goal of our work was to achieve such a cobalt doping, which we successfully did with a cobalt-to-iron ratio of 0.2, respectively $x \approx 0.5$.
2. In addition to the toxicity evaluations, it has already been shown in literature and also in our group that the incorporation of cobalt ions leads to an increased magnetocrystalline anisotropy of the particles compared to iron oxide. It is observed that an increase in the anisotropy constant (which consists of magnetocrystalline contributions) leads to a reduction in the average particle size that produces the highest hyperthermia performance. (<https://doi.org/10.1063/1.2830975>) As a result, cobalt ferrite particles should already be dominated by the Brownian relaxation mechanism in the high single-digit nanometer range (around 7 nm), compared to magnetite nanoparticles (around 14 nm). For example, an increase in the heating efficiency can be achieved by doping magnetite with cobalt to produce $\text{Co}_x\text{Fe}_{3-x}\text{O}_4$ within the concentration range of $x \leq 0.5$. (<https://doi.org/10.1016/j.physb.2020.412429>) Considering the different relaxation processes, the potential of the cobalt-doped systems is quite significant.
3. Additionally, realizing the silica-precursor-based synthesis could allow for further upscaling in hydrothermal reactors, as shown for precursor-derived hydrothermal methods (10.1039/D5RA02233A) by simultaneously enhancing the magnetic anisotropy, considering the future potential of this approach.

• Some figure captions are overly descriptive and lack conciseness, making it difficult to quickly grasp the key points.

We have tried to implement/reduce this as much as possible; however, due to the journal's guidelines, we are required to format figure captions in a way that their descriptions are comprehensible independently of the main text. The guidelines include:

- Figure captions must start with a brief title that describes the Figure as a whole and does not contain a reference to specific figure panels.
- Figure captions must also have a legend that defines each Figure panel individually.
- Readers should be able to understand the Figure without reference to the main text.
- Abbreviations, symbols, colors, and shading must be defined.
- The description of the statistical treatment of error analysis should be included in the figure legend.

- Section S7 on the mechanism of cobalt ferrite chain formation is truncated, limiting the understanding of how silica shells and PAA influence chain assembly. The MS should expand on this, possibly integrating a schematic diagram to illustrate the proposed mechanism.

We revised the order of the discussion on particle synthesis in the manuscript and added a figure to support the proposed mechanistic interpretation on page 24 - 28.

We also included a section discussing the influence of the PAA on particle synthesis (pg. 26, lines 491 - 499). Both the decreased size and size deviation value can be attributed to the specific interactions of additives, which primarily serve to inhibit excessive agglomeration of the seeds and particles. By enabling spatial separation, the number of nuclei increases, leading to a decrease in particle size. Based on this mechanistic understanding, the addition of PAA by modification of the akaganeite precursors is expected to significantly reduce the agglomeration of magnetite seeds, with nanorods acting as a source of iron ions during the particle growth process.

The influence of the silica shell on the particle formation is now presented in reference to Fig. 4. The particle chain assembly (pg. 28, lines 551 – 564) is also discussed. Since the chain formation, which is observable *via* TEM (though with limited conclusiveness), appears to increase with the silica shell of the precursor, this layer likely influences the process. Given that the pH value of 12 is established after the addition of NaOH to the reaction solution, the silica shell should gradually dissolve over time. (<https://link.springer.com/article/10.1007/s11051-023-05688-4>) This also explains the dissolution of the akaganeite particles. As the formation of the silica layer is a dynamic, base-catalyzed process, we assume that this occurs concurrently during the synthesis of cobalt ferrite particles. After the reaction, the pH value should be lowered due to the dissolution of silica species and the reacted chloride salts. Consequently, a reformation of the silica shell surrounding the magnetically interacting particles should occur. During the extended LaMer process, aggregation occurs spatially close to the particles, facilitating increased chain formation. If more silica is present, this process can be amplified, leading to enhanced chain formation, as the silica shells promote re-coating through hydrolysis and condensation reactions, as well as magnetic interactions between the simultaneously synthesized particles.

- The effect of temperature on cobalt ferrite phase purity is described, but the underlying chemical or thermodynamic factors are not addressed.

Based on the proposed mechanism, cautious statements regarding the influence of reaction temperature have been added on pg. 16, lines 314-320 and pg. 18, lines 358-371.

Comparison of the magnetic particle shapes synthesized with aka and aka@PAA shows a more angular morphology at low temperatures, here 160°C. In contrast, at high temperatures, at 190 °C, the particles tend to become more rounded and appear quasi-spherical. However, it is particularly well known that Ostwald ripening predominantly occurs at the edges and corners of cubic and octahedral particles. Consequently, the morphology changes to significantly truncated particles with higher temperatures, as can be observed in the TEM images (Fig. 3).

The discussion of the temperature effect will be based on the mean values of both trials. The particle fractions produced using silica precursors suggest that the lower temperature favor the formation of smaller sizes. This could be related to the dissolution of silica structures under alkaline conditions, with the dissolution rate depending on the thickness of the silica layer (<https://link.springer.com/article/10.1007/s11051-023-05688-4>). The amount of dissolved silica increases at high temperatures, due to an enhanced dissolution rate and higher saturation concentration of silica during the reaction in water. (<https://link.springer.com/article/10.1007/s11051-023-05688-4>) At high temperatures, coupled with

increased amounts of dissolved silica, there is less spatial separation of nuclei during the particle growth process, allowing for the formation of larger particle diameters. Conversely, increased salinity of the solution enhances the stability of the silica shells, which may preferentially occur at lower temperatures and the associated lower kinetic energy in the system. This would result in a higher proportion of ionic compounds, such as unreacted Fe^{3+} , Fe^{2+} , and Co^{2+} , remaining in the aqueous solution with a lower activity coefficient of water over a longer period of time. The silica layers would still be present in the solution and support the formation of magnetic particles close to these silica shells, thereby favoring the development of more chain-like structures.

- The SI reports large standard deviations for some measurements (e.g., ($\sigma_{\text{TEM}} = 10.8$) nm for CF-S3-1, SI, p. 6), indicating high polydispersity, but the MS does not discuss implications for practical applications.

We implemented a section on potential biomedical applications, also regarding the size distribution on pg. 49, lines 957 - 964. It has been demonstrated that increased polydispersity in particle size has an impact on hyperthermia, typically resulting in a decrease in efficiency. The particles synthesized here exhibit size standard deviations ranging from 18% to 35%. Therefore, the dispersity is calculated, resulting in values ranging from 0.04 to 0.13, except the outliers CF-S1-1 and CF-S3-1 (see Supplementary Tab. S4.2). It should first be noted that these values are comparable to those reported for aqueous coprecipitation reactions (10 – 30% or PDI = 0.01–0.09, respectively), even if they appear relatively high in absolute terms in this work. Furthermore, the use of PAA or silica in the reaction solution does not result in a clear trend in dispersity. However, an average silica thickness of around 4 nm appears to have a positive influence on dispersity, resulting in the lowest PDI.

- Some TEM images show empty silica shells which gives the impression of inhomogeneous and uncontrolled silanization of the aka@PAA. Therefore, it would be great to see EDX mapping to properly see the localization of akaganeite in the SiO_2 -coated aka@PAA. Also, EDX mapping of CF nanoparticles to show the localization of Co and Fe is recommended.

We thank reviewer #3 for the suggestion. We would rather doubt the occurrence of empty shells, since thermodynamically, empty silica particles would predominantly form spheres. Our core-shell structures, which, due to the anisotropic nature of the used akaganeite, should also adopt nanorod-like shapes. However, we conducted additional HR-TEM, line-scan profiles, and EDX measurements of the precursor systems. For this purpose, we analyzed two samples with 2.6 nm and 5.6 nm and revised Fig. 2 accordingly. To demonstrate the presence of akaganeite in the core of the silica precursors and to evaluate the porosity of the silica layers, two fractions were analyzed with $d_{\text{SiO}_2} < 4$ nm and $d_{\text{SiO}_2} > 4$ nm (see Supplementary Sect. S3.2). In the HR-TEM images, a clear distinction between akaganeite and silica can be observed (Fig. 2h and Fig. 2j, see Supplementary Fig. S3.4). Additionally, Energy Dispersive X-ray spectroscopy (EDX) analysis reveals silicon signals at the surface and edges of the particle, whereas iron is present primarily in the core, especially for the sample with the thicker shell. The transition is clearly visible in the line-scan profiles (Fig. 2k, see Supplementary Fig. S3.6). However, in the sample with the thin silica shell, where the line scans do not appear to be very conclusive (Fig. 2i, see Supplementary Fig. S3.5), a Si/Fe ratio of 0.28 is observed based on the distribution in the sum spectra (see Supplementary Fig. S3.7). This is significantly different from the Si/Fe ratio of 0.09 that was determined in the bare akaganeite sample (see Supplementary Fig. S1.2). In the sample with a thicker shell, the silicon content correspondingly increases, resulting in a Si/Fe ratio of 2.7 (see Supplementary Fig. S3.7). Chloride atoms, located within the tunnel structures of akaganeite, can also be detected. This confirms that akaganeite remains stable and is neither dissolved nor transformed into another phase during the PAA functionalization and the Stöber process. We implemented this discussion on pg. 14, line 265.

- The MS focuses heavily on synthesis and characterization but does not adequately discuss how the chain-like arrangements enhance magneto-responsive properties compared to isotropic nanoparticles. For example, how do the chains improve magnetic anisotropy or coercivity?

We have reviewed potential applications concerning the following works, as they primarily address enhanced magnetic and shape anisotropy, reversibility through external fields, increased (rough) surfaces, magnetic coupling effects, and other properties of bioinspired magnetic nanochains. In particular, we would like to refer to the studies as follows,

- <https://doi.org/10.1021/acsami.0c03220>,
- <https://doi.org/10.1002/adma.201707515>,
- <https://doi.org/10.1002/adv.202309564>,
- <https://doi.org/10.1039/D4NR02643H>,
- <https://doi.org/10.1038/s41467-018-04172-1>, and
- <https://doi.org/10.1186/s40824-022-00278-2>,

which focus on the construction of silica-coated, at least magnetically-responsive nanochain structures and demonstrate their applications across the various fields mentioned in the introduction. We implemented them in the introduction on pg. 4, lines 64 – 69. We additionally added a section regarding the influence of chain assemblies on pg. 51, lines 1022 – 1062.

- Potential applications are mentioned briefly in the introduction but not revisited in the discussion. A section on how the findings could impact fields like magnetic hyperthermia or other field would strengthen the manuscript.

We have included a comprehensive section for a cautious classification and description of the particle systems, focusing on their SAR and potential for biomedical applications on pg. 48 – 55. In this context, the coercivities, saturation magnetizations, and remanence values determined in Fig. 5 were also considered in this discussion, along with their size and size distribution. In the context of toxicity, cellular distribution, and biodegradation of cobalt ferrite and cobalt-doped ferrite nanoparticles, a link was established between the literature and our particles regarding their size, magnetization, and chain arrangements. Despite the controversial discussion, it is worth noting that cobalt ferrite particles of smaller sizes can achieve comparable SAR values to those of magnetite. These small particles are expected to evade the immune system more effectively and have a longer circulation time in blood vessels, thereby increasing the likelihood of reaching the tumor tissue. Additionally, for magnetic carriers, small nanoparticles are more stable against aggregation, which prevents precipitation and consequently reduces the risk of blood vessel occlusion. This combination of high SAR performance and enhanced biological stability underscores the importance of nanoscale cobalt ferrite particles for biomedical applications. In conclusion, several studies have discussed the toxicity of cobalt ferrite and cobalt-doped ferrite particles, which are found to be size- and concentration-dependent. Interestingly, some refer to toxicity as a side effect and a factor to consider, with biocompatibility being enhanced through silica or polymer coatings. On the other hand, some approaches utilize the minimal release of Co (and Fe) for concurrent chemotherapy, since thermotherapies alone in most cases do not result in significant tumor reduction.

- The MS and SI contain minor typographical errors, such as "akageneite" instead of "akaganeite".

We revised the manuscript and SI accordingly.